# Dynamic weakening during earthquakes controlled by fluid thermodynamics

M. Acosta[1], F.X. Passelègue[1,2], A. Schubnel[3] & M. Violay [1]

Earthquakes result from weakening of faults (transient decrease in friction) during co-seismic slip. Dry faults weaken due to degradation of fault asperities by frictional heating (e.g. flash heating). In the presence of fluids, theoretical models predict faults to weaken by thermal pressurization of fault fluid. However, experimental evidence of rock/fluid interactions during dynamic rupture under realistic stress conditions remains poorly documented. Here we demonstrate that the relative contribution of thermal pressurization and flash heating to fault weakening depends on fluid thermodynamic properties. Our dynamic records of laboratory earthquakes demonstrate that flash heating drives strength loss under dry and low (1 MPa) fluid pressure conditions. Conversely, flash heating is inhibited at high fluid pressure (25 MPa) because water's liquid–supercritical phase transition buffers frictional heat. Our results are supported by flash-heating theory modified for pressurized fluids and by numerical modelling of thermal pressurization. The heat buffer effect has maximum efficiency at mid-crustal depths (~2–5 km), where many anthropogenic earthquakes nucleate.

[1] Laboratory of Experimental Rock Mechanics (LEMR), École Polytechnique Fédérale de Lausanne (EPFL), Station 18, CH-1015 Lausanne, Switzerland. [2] School of Earth and Environmental Sciences, University of Manchester, Manchester M13 9PL, UK. [3] Laboratoire de Géologie, CNRS UMR 8538, École Normale Supérieure, 75005 Paris, France. Correspondence and requests for materials should be addressed to M.A. (email: mateo.acosta@epfl.ch)

During earthquakes, fault zones often saturated with fluids are sheared over several metres at slip rates of metre per second, under normal stresses up to hundreds of megapascal, generating very high frictional power per unit area[1]. This large power triggers thermally activated weakening mechanisms (flash heating (FH) and thermal pressurization (TP)) responsible for low dynamic friction[2–18].

Theoretical models predict that TP dominates at large slips and/or above mid-crustal depths, while FH is the dominant weakening mechanism at small slips and/or greater depths[12]. However, these models rarely incorporate water thermodynamics (notably phase transitions). Moreover, in contrast to FH, TP during seismic slip has not been established experimentally as of yet[8,17], due to the difficulty of reproducing spontaneous dynamic rupture under realistic stress and fluid pressure conditions in the laboratory. In summary, fully controlled experiments studying dynamic shear instabilities in the presence of fluid pressures (pf) have been lacking so far and hydro-thermo-mechanical couplings during dynamic rupture remain still unclear.

Here we conducted stick-slip experiments (laboratory proxies for earthquakes[19]) on Westerly Granite (WG) saw-cut samples (Supplementary Fig. 1) under triaxial stress conditions (principal stresses $\sigma_1 > \sigma_2 = \sigma_3$). The experiments were done at stresses representative of the upper continental crust[19] (effective confining pressures $\sigma_3' = \sigma_3 - pf = 70$ MPa). We imposed three different fluid pressure levels: dry, low fluid pressure and high fluid pressure, which correspond to pf = 0, 1, and 25 MPa, respectively (hereafter referred to as dry, $Low_{Pf}$ and $High_{Pf}$, respectively). Combining dynamic stress evolutions with on-fault resolved displacements and microstructural analysis of the postmortem specimens evidenced that distinct dynamic weakening mechanisms (FH and TP) were activated at the different fluid pressure levels. Further, we applied thermal weakening models to our experimental data including the evolution of fluids thermophysical properties with pressure and temperature. The results showed that both FH and TP were activated during co-seismic slip and that their relative contributions are controlled by the evolution of water's thermophysical properties at phase transitions.

## Results and discussion

**Mechanical results.** Continuous records of shear stress ($\tau$) versus time (Fig. 1a) showed that, during a stick-slip event, $\tau$ dropped from an initial peak static value ($\tau_0$) down to a final residual value ($\tau_f$), resulting in a static stress drop ($\Delta\tau_s = \tau_0 - \tau_f$). High-frequency records[9,10] (Fig. 1b–d) showed that $\tau$ first increased from $\tau_0$ up to a peak dynamic value $\tau_p$ and then abruptly dropped to a minimum dynamic value $\tau_{min}$ before recovering to $\tau_f$, thereby defining a dynamic (or breakdown) stress drop ($\Delta\tau_b = \tau_p - \tau_{min}$). The dynamic rise of $\tau$ up to $\tau_p$ resulted from stress amplification (i.e. stress intensity factor) at the rupture tip[20].

Earthquakes during dry experiments presented larger $\Delta\tau_s$ values (Fig. 2a) (from 30 to 45 MPa versus from 10 to 30 MPa and from 5 to 18 MPa under $Low_{Pf}$ and $High_{Pf}$, respectively), while larger breakdown stress drops were recorded under $Low_{Pf}$. There $\Delta\tau_b$ was 14% larger on average than in dry conditions and was remarkably 73% higher than at $High_{Pf}$. Note that $\tau_0$ (i.e. the amount of elastic energy stored in the system) was similar at both fluid pressures but smaller than in dry conditions. The total slip per event was similar for dry and $Low_{Pf}$ conditions but was two times smaller under $High_{Pf}$. In all conditions, peak static frictional strength ($\mu_0 = \tau_0/\sigma_{n0}'$) ranged between 0.6 and 0.89 (Fig. 2b, compatible with Byerlee's law[21]) but was approximately 17% higher in dry experiments than at $Low_{Pf}$ and $High_{Pf}$. This indicates lower static shear strengths in the presence of fluids that resulted from a reduction of adhesion along fault surface in the presence of water[22] (i.e. to a decrease of the contact surface energy). Regarding weakening processes, the dynamic friction ($\mu_d = \tau_{min}/\sigma_n'$, see Methods) was lower at $Low_{Pf}$ (from 0.02 to 0.24) than at dry (from 0.29 to 0.39) conditions and $High_{Pf}$ (from 0.42 to 0.51). Such differences in the dynamic fault strength (i.e. evolution of $\tau$ and $\mu_d$) in these three experiments (performed at constant $\sigma_3'$) suggest the activation of distinct weakening mechanisms during earthquake rupture. Such mechanisms seem less effective at $High_{Pf}$ (i.e. smaller slip and higher $\mu_d$ for equivalent $\tau_0$ and $\Delta\tau_s$) and slightly more effective at $Low_{Pf}$ (i.e. larger $\Delta\tau_b$ and lower $\mu_d$, leading to a transient, almost total strength loss).

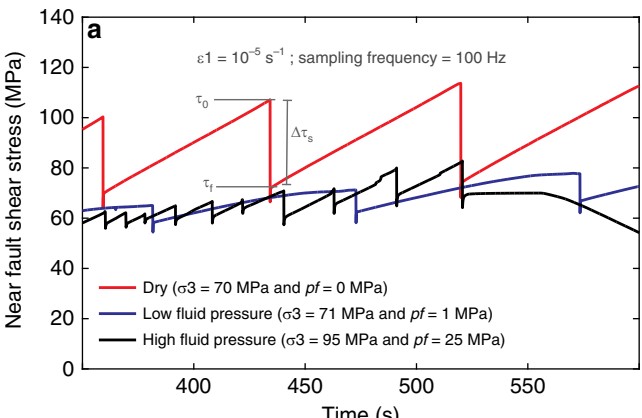

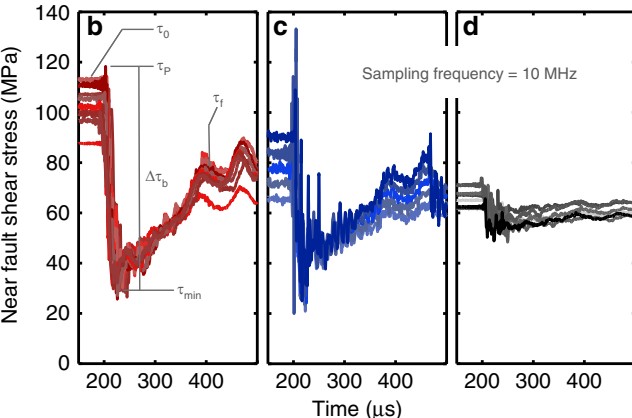

**Fig. 1** Mechanical results for experiments at $\sigma_3' = 70$ MPa. **a** Near-fault shear stress evolution with time. Static stress drop ($\Delta\tau_s = \tau_0 - \tau_f$) is shown as an example. **b–d** Dynamic shear stress evolution with time. Each curve corresponds to one stick-slip event, change in colour hue accounts for different events. In addition to $\tau_0$ and $\tau_f$, the maximum and minimum dynamic values of shear stress, $\tau_p$ and $\tau_{min}$, are presented as examples, defining a breakdown stress drop ($\Delta\tau_b = \tau_p - \tau_{min}$). **b** Dry experiment, red curves. **c** Low fluid pressure experiment, blue curves. **d** High fluid pressure experiment, black/grey curves

**Microstructural observations.** Scanning electron microscopy on postmortem fault surfaces (Fig. 3, Supplementary Fig. 4) revealed ~20 μm long patches of ropey, stretched material elongated along the shear sense in both the dry (Fig. 3b) and $Low_{Pf}$ (Fig. 3c) experiments. These structures are consistent with melting of fault asperities during co-seismic slip and may explain the strong weakening observed in these experiments[9,10,18]. Conversely, such structures were not found at $High_{Pf}$ (Fig. 3d) where the surfaces

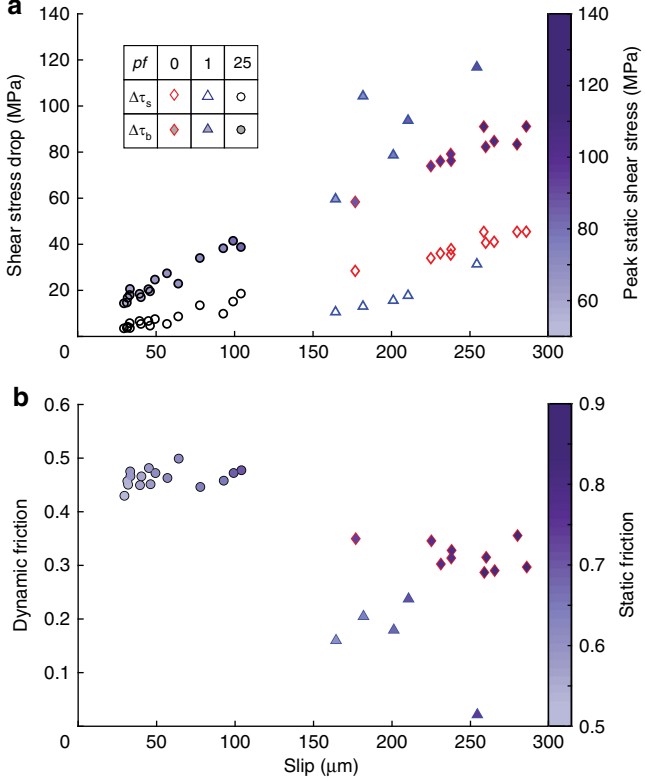

**Fig. 2** Compiled data for experiments at $\sigma_3' = 70$ MPa. Red diamonds account for dry events, blue triangles for low fluid pressure events and black circles for high fluid pressure events. **a** Compiled data of $\Delta\tau_s$ (static shear stress drop, empty markers) and $\Delta\tau_b$ (breakdown shear stress drop, full markers) against on-fault resolved slip. Colour bar accounts for $\tau_0$ (static shear stress reached at the onset of the instability, i.e. amount of elastic energy stored in the system). **b** Dynamic friction resolved against the on-fault total event slip. Colour bar accounts for static friction reached at the onset of instability

were covered with asperity debris of sizes ~0.2–5 μm. No evidence of melt was found, suggesting that lower temperatures were reached at the asperity contacts and confirming the reduced efficiency of frictional heating and weakening at High$_{pf}$.

**Flash temperature in the presence of pressurized fluids.** To support our experimental and microstructural results, we computed the flash temperatures (maximal transient temperatures) reached on 20 μm radius asperities due to shear heating[2,23] during their lifetimes ($t_c$) (Methods, Fig. 4a). In the presence of fluids, water cools asperities through heat capacity and latent heat (acting as a heat barrier) of a finite water volume surrounding the highly stressed asperity[7] (Fig. 4b). The main hypothesis of the model is that the fluid volume surrounding the asperities is at thermal equilibrium with the asperity. This assumption should remain valid during frictional slip since the thermal diffusion length ($\sqrt{\pi.\kappa_{th}.t}$ with $\kappa_{th}$ the thermal diffusivity and $t$ the heating time) is close to the asperity size at FH velocities (commonly admitted ~10 cm s$^{-1}$ (refs. [2–7])), and when the solid–solid contact starts slipping, a liquid–solid contact forms immediately, allowing for fast temperature equilibrium between the asperity and the surrounding fluid. Conversely to previous studies[7,24], we also included the isobaric evolution of water's specific and latent heat ($c_{pw}$ and $L_w$), as well as density ($\rho_w$) with temperature in the calculation[25] (Fig. 4c–e, Methods, Supplementary Fig.7). Given

our experimental conditions, we considered water cooling of asperities as a purely diffusive mechanism (no advection) for fault permeabilities <10$^{-17}$ m$^2$ at Low$_{pf}$ and <10$^{-18}$ m$^2$ at High$_{pf}$ (Methods, Supplementary Fig. 5). Note that in this model we considered the maximum temperature that can be reached at asperities affected by the cooling effect of water[7]. Such temperature differs from the temperature history at asperities during slip. Under dry conditions, when no buffering takes place, the flash temperature becomes the exact solution for the one-dimensional heat diffusion problem[2,10,26] at the asperity scale (assuming the contact shear stress at asperities rather than the macroscopic shear stress distributed along the interface). There temperature increased as a power law of slip (see Supplementary Fig. 6 for other asperity sizes). The FH temperature (approximately 1000 °C[2–5]) was reached for slip rates >10 cm s$^{-1}$ during the asperity lifetime, as predicted by FH theories and previous experiments[2–7]. At those velocities, in the Low$_{Pf}$ case, water-buffered temperatures were observed in the first half of the contact lifetime, and so, flash temperatures remained <179 °C, i.e. while water stayed in a liquid state. Longer slips at such seismic velocities (e.g. higher frictional heat) allowed water to overcome the liquid–vapour phase transition temperature during $t_c$, inducing a strong drop in $\rho_w$ and $c_{pw}$ (roughly falling to 0.5% and 50% of their room temperature values, respectively; Fig. 4c, d). In this case, vaporization of water enhanced shear heating at contacts and allowed FH of asperities for slip velocities larger than ~10 cm s$^{-1}$, as also observed in dry conditions. Conversely, at fluid pressures ranging from 25 to 70 MPa, temperature rise was strongly buffered by water cooling during $t_c$ due to the liquid–supercritical transition. This phase change requires a distributed amount of energy over a finite temperature range, opposed to the case of isothermal vaporization where $L_w$ acts as a heat barrier. Therefore, the heat capacity of water increases by 1400% during the transition at pf = 25 MPa (Fig. 4d) while the drop in density is smoother than in the case of vaporization[25] (Fig. 4c). At high fluid pressures, water turned out to be an extremely efficient heat buffer, inhibiting FH phenomena and hindering rises in temperature to the liquid–supercritical phase transition temperature (~373 °C at pf = 25 MPa, Supplementary Fig. 7) at asperity contacts during their lifetime. Temperature rise was buffered even for slip rates of 1 m s$^{-1}$ (admitted slip rate during earthquake propagation[1,2,4,6]). This major heat sink explains the reduced dynamic weakening observed at High$_{Pf}$ and the absence of frictional melt on the fault surfaces.

**Shear heating and TP of fluid saturated faults.** The liquid–vapour transition has been thought to have strong thermal effects on faulting, inhibiting temperature rise due to TP during co-seismic slip[14,15]. In high-velocity friction experiments[8,14], TP enhanced the friction drop of ~0.1, which is comparable to the difference observed between the dynamic friction recorded during Low$_{Pf}$ and dry conditions (Fig. 2b). Such difference could also be due to a reduction of melt viscosity through hydration in the presence of fluids. However, rotary shear experiments have demonstrated that the chemical compositions of melts developed after long slip times (>10 s) under vacuum, room humidity and fluid-saturated conditions were identical[7], discarding the possibility of melt-hydration in our experiments (here the total slip time was <30 μs). TP due to fluid pressurization could then be a candidate to explain the slightly lower dynamic friction values observed at Low$_{pf}$ while FH remains the dominant weakening mechanism.

While FH explains the large dynamic strength drops observed in dry and Low$_{Pf}$ conditions, it does not explain the small strength drops observed at High$_{Pf}$ conditions. To explain the

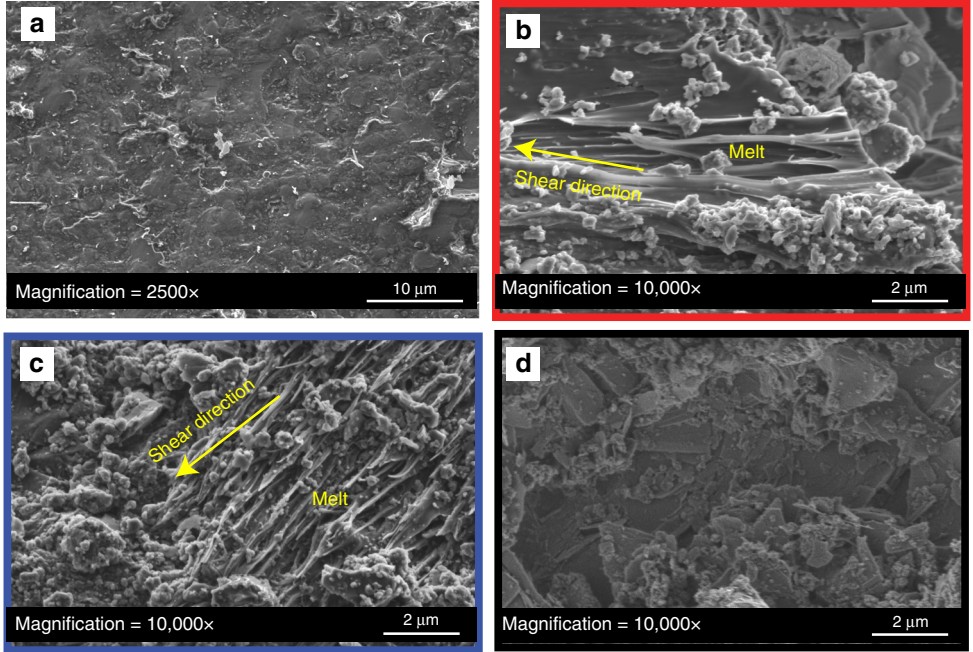

**Fig. 3** Micrographs of the fault's surfaces. Scanning electron microscopic images under secondary electron mode of half-fault surfaces. **a** Initial sample. Homogeneous surfaces with asperities of ~2–20 μm. **b** Surface of the dry experiment. Surfaces were covered with clasts of sizes <5 μm. Magnification allowed observation of melted structures stretched along the shear direction. **c** Surface of the low fluid pressure experiment. Clasts of sizes <5 μm and melted patches were also identified. **d** Surface of the high fluid pressure experiment. Clasts of sizes <5 μm were found all over the surface. No melted structures were found

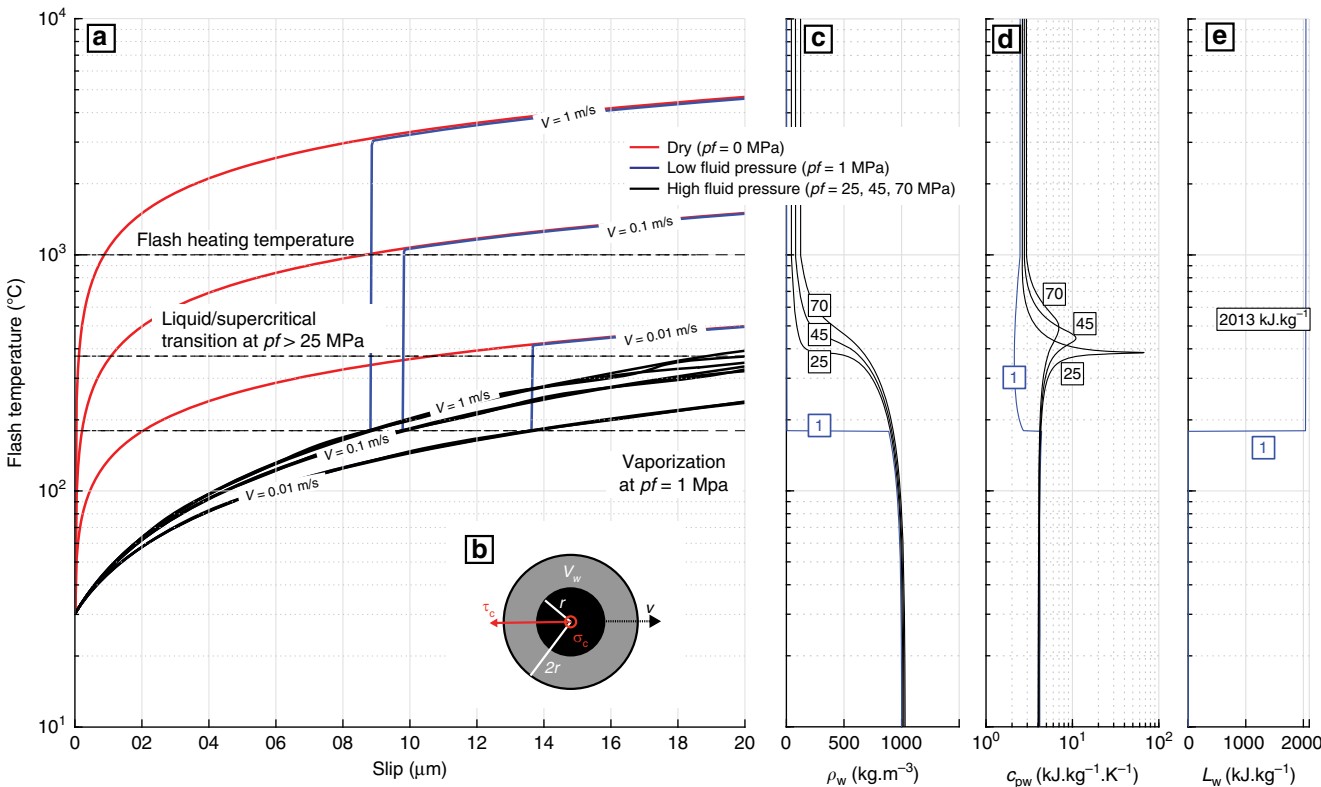

**Fig. 4** Flash temperature calculation. Red curves correspond to dry conditions, blue to low fluid pressure (1 MPa) and black to high fluid pressures (25, 45 and 70 MPa). **a** Flash temperature reached at the asperity contacts versus slip (e.g. Methods). **b** Schematic top view of the considered contact geometry, in black we see the asperity contact of radius $r = 20$ μm stressed at a normal stress $\sigma_c$ and at a shear stress $\tau_c$. The asperity is surrounded by a volume of water $V_w$, which buffers temperature. **c** Temperature versus water density[25] for the different fluid pressures in MPa. **d** Temperature versus water specific heat[25]. **e** Temperature versus water's latent heat[25]. Note that the latent heat ($L_w$) decreases with rising pressure until it vanishes at the critical point (~22 MPa and ~373 °C)

   

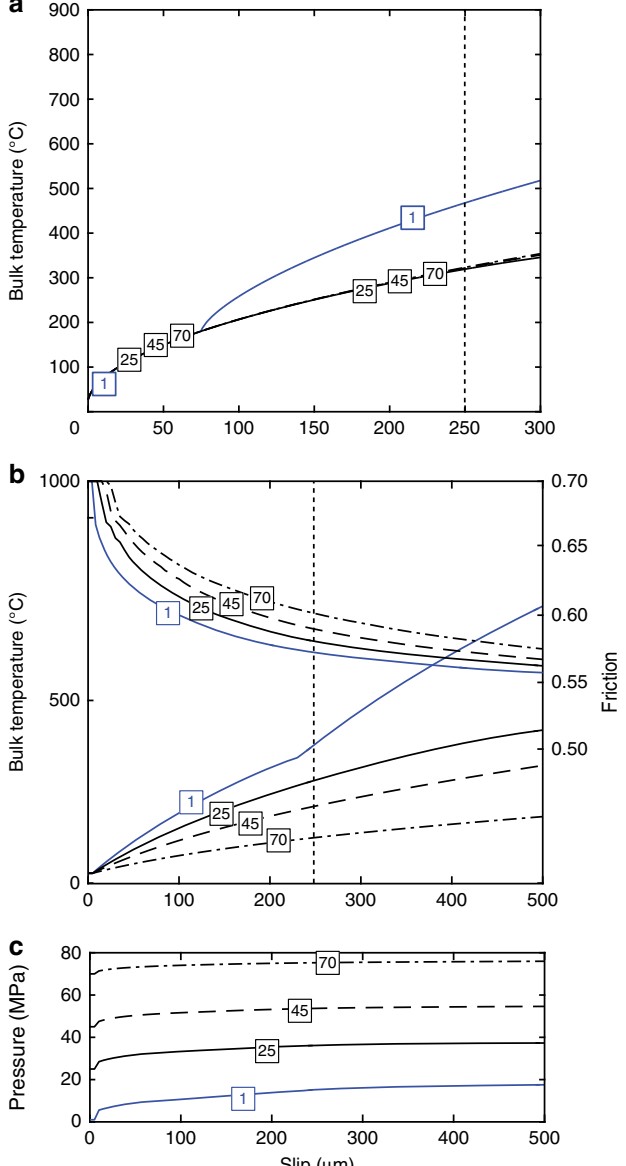

**Fig. 5** Bulk fault temperature model at 1 m s$^{-1}$ slip rate under experimental shear loading. Blue curves correspond to low fluid pressure conditions (1 MPa) and black to high fluid pressures (25, 45 and 70 MPa). Vertical dotted lines show the maximum slip in our experiments. **a** Bulk temperature reached in the fault's slipping zone during shear heating versus slip under drained conditions (e.g. Methods); each curve corresponds to the labelled fluid pressure in MPa. **b** Bulk temperature (left y axis) reached in the fault during shear heating under undrained conditions and the corresponding friction evolution due to TP (right y axis) versus slip (e.g. Methods); each curve corresponds to the labelled fluid pressure in MPa. **c** Fluid pressure evolution in the fault's slipping zone during shear heating versus slip for the different initial fluid pressures labelled in MPa

small stress drops observed at High$_{Pf}$, we computed the temperature evolution on a bulk planar fault in both drained (Fig. 5a) and undrained (Fig. 5b, c) conditions using a finite difference numerical model (e.g. Methods). We considered full thermodynamic evolution of fluid properties with pressure and temperature[15,16,25]. Under drained conditions, we observed that the reached temperatures (which are a maximum estimation of the possible temperature in the bulk fault since the shear stress for

heat generation is taken as the static fault's shear strength of our experiments) remained below rock's thermal degradation temperature (~1000 °C) even for slips larger than the maximum slip observed in the experiments (~250 μm) (Fig. 5a). This observation is in agreement with our microstructural observations, since melting was not pervasive over the sample surface but was localized at asperity scale (Fig. 3c and Supplementary Fig. 4c), as predicted by FH theory. Nevertheless, under our experimental conditions, the water heat buffer effect due to the liquid–supercritical transition was still observed for initial pore pressures >22 MPa. Note that strong temperature rises on the fault due to instantaneous water vaporization took place in a similar manner than for the flash temperature computations, confirming our calculations at the asperity scale. Under undrained conditions (Fig. 5b, c), since the fault's stress obeys the effective pressure law, we observed an initial fast decay in friction due to TP. The decay then stabilized leading to friction drops of ~0.1–0.15 for slips of ~20–150 μm in all fluid pressure conditions. Such friction drop values are remarkably consistent with the friction drops observed in High$_{Pf}$ experiments (Fig. 2b). Therefore, at High$_{Pf}$, TP might well be the dominant weakening mechanism in our experiments, as supported by our microstructural analysis.

**Implications for natural and induced earthquakes**. Similar stress evolutions observed in experiments conducted at other effective stresses and at pf = 45 MPa (Supplementary Fig. 3) suggest that the observed heat buffer operates even at higher fluid pressures, where the liquid–supercritical transition is smoother[25] (e.g. Fig. 4c, d and Supplementary Fig. 7). To further study the depth dependence of this heat buffer effect, we computed again the temperature rises (in both drained (Fig. 6a) and undrained conditions (Fig. 6b, c)) due to TP with a depth extrapolation. The extrapolation was done for a mean stress equal to the lithostatic overburden gradient of 27 MPa km$^{-1}$, a hydrostatic fluid pressure rise of 10 MPa km$^{-1}$, a geothermal gradient of 30 °C km$^{-1}$ (ref. [27]) and an initial friction of 0.7 (e.g. Methods). We observed that a heat buffer can operate for fluid pressures up to 45 MPa in both drained and undrained cases but its efficiency is strongly reduced when fluid pressures reach 70 MPa (~7 km depth). At large depths, higher background stress and a smoother super-critical transition allow to overcome the transition temperature for smaller slips when sliding at seismic slip rates (~1 m s$^{-1}$), consistent with previous studies on the depth dependence of weakening mechanisms[12]. Nevertheless, the dynamic friction values predicted by TP theory are similar at all depths for a given final slip, likely because at greater depths (>7 km), the background driving stress has a stronger effect than the pore fluid pressure rise on TP at small slip[12].

Previous TP models considered the liquid–supercritical transition and found no significant effect of the transition on dynamic ruptures[16] but did not consider the effect of FH at the microscopic level. Here we demonstrate that water phase transitions may control FH at the asperity level by acting as a major heat buffer and so they can control earthquake rupture. The initial fluid pressure level is a critical parameter that cannot be neglected via the effective pressure concept because it controls water thermodynamics. Thus dependencies on temperature and pressure of thermodynamic fluid properties should be taken into account in future weakening models, in particular at the microscopic level[2,12,13]. Extrapolation of our results to natural conditions suggests that the heat buffer effect has a maximum efficiency at mid-crustal depths (~2–5 km) where major anthropogenic earthquakes appear[28].

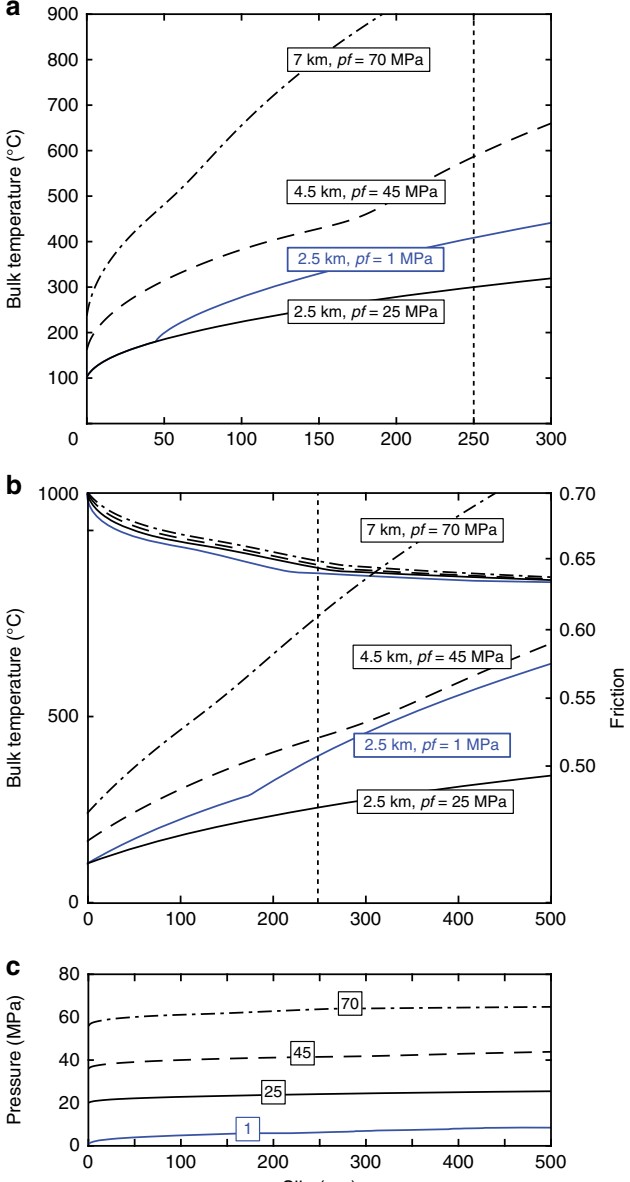

**Fig. 6** Bulk fault temperature at 1 m s$^{-1}$ slip rate, at upper crustal depths. Blue curves correspond to low fluid pressure conditions (1 MPa) and black to high fluid pressures (25, 45 and 70 MPa). Vertical dotted lines show the maximum slip in our experiments. **a** Bulk temperature reached in the fault during shear heating under drained conditions versus slip (e.g. Methods). **b** Bulk temperature (left y axis) reached in the fault during shear heating under undrained conditions and the corresponding friction evolution due to TP (right y axis) versus slip (e.g. Methods); each curve corresponds to the labelled depth. **c** Pore pressure evolution in the fault during shear heating versus slip for the different initial fluid pressures

## Methods

**Starting material.** Samples were WG cylinders (Supplementary Fig. 1) (40 mm diameter and 88 mm length). This material was selected because it is simple in composition, it is representative of the continental crust and because of its very fine grain (<1 mm), perfect homogeneity, isotropy and low alteration degree. A thermal treatment at atmospheric pressure was performed on the samples in order to increase the samples' crack density (i.e. permeability) and so to allow good saturation and reasonable fluid diffusion times in the samples. Samples were heated at a gradient of 5 °C min$^{-1}$ up to 450 °C. Then the target temperature was kept constant for 2 h. Finally, the furnace was turn off and the samples were let to cool down overnight. The temperature ramp imposed in the furnace allowed having no

temperature gradient inside the sample during heating. Since the thermal diffusivity of the rock is $\sim\kappa_{th} = 10^{-6}$ m$^2$ s$^{-1}$, and with a sample radius of $r = 2 \times 10^{-2}$ m, temperature should equilibrate in $r^2/\kappa_{th} = 400$ s (~7 min). Thermal cracking arose from differential thermal expansion of neighbouring grains thus allowing increasing both intergranular and grain boundary microcracking without overcoming the Quartz Alpha-Beta transition (578 °C). Permeability was measured after thermal treatment and was ~5 times higher than that of untreated samples (from $2 \times 10^{-19}$ m$^2$ to $1 \times 10^{-18}$ m$^2$ at 5 MPa confining pressure). The cylinders were then cut to the correct length, and the top and bottom bases grinded to ensure perfect planarity with the horizontal. Then the samples were saw-cut at an angle ($\theta$) of 30° to the sample's long axis to create an artificial elliptical fault of major axis $L = 80$ mm and minor axis $l = 40$ mm. The apparent contact area being: $A = \pi * \left(\frac{L}{2} * \frac{l}{2}\right) = 2513$ mm$^2$. Fault's surfaces were then grinded to ensure perfect contact in the fault and then roughened with #240 grit paper to ensure a minimum cohesion along the fault's interface and impose a constant fault roughness in all the specimens.

**Experimental set-up.** The apparatus used was the tri-axial press of ENS Paris built by Sanchez Technologies. It is a servo-controlled oil medium confining cell with maximum confining pressure of 100 MPa. Axial loading was controlled by a separated servo pump acting on an axial piston (maximal stress of 680 MPa on 40 mm diameter samples). Fluid pressure regulation was assured by a double syringe pump (Quizix 20k) able to reach 120 MPa fluid pressures (1 kPa pressure accuracy, 1 μL volume accuracy). Under this configuration, shear stress ($\tau$), normal stress ($\sigma_n$) and slip ($D_f$) resolved on the fault can be expressed as:

$$\tau = \left(\frac{\sigma_1' - \sigma_3'}{2}\right)\sin(2(90 - \theta)) \quad (1)$$

$$\sigma_n' = \left(\frac{\sigma_1' + \sigma_3'}{2}\right) + \left(\frac{\sigma_1' - \sigma_3'}{2}\right)\cos(2(90 - \theta))$$

where $\sigma'$ refers to effective stress as $\sigma' = \sigma - pf$
and

$$D_f = \frac{D_1}{\cos(\theta)} = \frac{(\varepsilon_{1s}).L}{\cos(\theta)} = \frac{\left((\varepsilon_{1ext}) - \left(\frac{\Delta\sigma}{E_{ap}}\right)\right).L}{\cos(\theta)} \quad (2)$$

where $\varepsilon_{1ext}$ is the measured axial strain on the whole system; $\varepsilon_{1s}$ is the axial strain of the sample corrected by the stiffness of the apparatus using linear elasticity, $\Delta\sigma = (\sigma_1 - \sigma_3)$ the differential stress; $E_{ap}$ the stiffness of the apparatus; $L$ the sample's length; $D_1$ the axial displacement of the sample and, finally, $D_f$ the projected displacement on the fault.

Finally, making the reasonable assumption that confining pressure ($\sigma_3$) does not change during stick-slip events, near fault friction ($\mu$) is calculated as:

$$\mu = \frac{(\sigma_1' - \sigma_3')\sin(2(90 - \theta))}{(\sigma_1' - \sigma_3')(\cos(2(90 - \theta)) + 1) + 2\sigma_3'} \quad (3)$$

The recorded parameters during deformation were as follows. In the far-field, we recorded the axial and confining pressures through pressure transducers of 0.001 MPa resolution. In addition, axial displacement was measured by recording three Foucault current sensors of 0.1 μm resolution. The sampling rate on far field sensors was 100 Hz. These provided the macroscopic deformation of the system (sample plus apparatus deformation). In the near-field, we measured stress and strain through strain gages glued 3 mm away from the fault (Supplementary Fig. 1). Gages were coated with a cyanoacrylate gel, which prevented shortages due to pressurized water. These sensors allowed a local recording of the principal strains at 10 MHz sampling frequency. A full (Wheatstone) bridge configuration gage (HBM 3/350 VY41) allowed measuring directly the differential strain ($\varepsilon_1 - \varepsilon_3$). To calibrate the gage, we measured the constant Young modulus of the rock during elastic loading phase. Then we had direct conversion from the strain recorded at the gage to the corresponding far field measured stress using the measured sample's Young modulus (differential stress ($\sigma_1 - \sigma_3$)) (Supplementary Fig. 2). Gages allowed to record the dynamic stress change of each stick-slip event through an acoustic emission trigger set-up[9,10] at 10 MHz sampling frequency. Strain gage data were recorded continuously at 100 Hz to observe the overall evolution of near fault shear stress.

**Loading procedure and laboratory earthquakes.** Stick slip experiments were performed under nominally dry and fluid pressure conditions. Confining pressures ranged from 50 to 95 MPa, and fluid pressure from 0 (i.e. dry) to 45 MPa. Constant strain rate was imposed at ~1 × 10$^{-5}$ s$^{-1}$ (see Supplementary Table 1 for the detailed experimental matrix). The experimental procedure was as follows: We first increased the confining and axial pressures up to 10 MPa. Then, in case of pressurized fluid experiments, we carefully flushed air away from the sample, then we increased the fluid pressure to 5 MPa in both upper and lower reservoirs. We then waited for pressure and volume equilibrium between the two reservoirs. The axial,

confining and fluid pressures were increased (fluid pressure was decreased in low fluid pressure experiments) in parallel up to their target values. Again, we waited for fluid pressure and volume equilibrium between the reservoirs. Finally, axial pressure was increased at constant axial loading rate while fluid and confining pressures were held constant. Both shear and normal stresses increased with axial loading until shear stress reached the strength of the fault. At this point, the stick-slip instability occurred and was accompanied by a brutal release of shear stress and seismic slip on the fault plane. Such an event is a stick-slip event or laboratory earthquake.

The reproducibility of the shear stress evolution for successive events in each configuration suggests that the possible change in surface topography with increasing number of events did not affect the fundamental processes accounting for earthquake rupture propagation.

**Flash temperature computation.** Flash temperature is the maximum transient temperature responsible for fast weakening of fault frictional strength during sliding[23]. Flash temperature is reached at an asperity for the lifetime of contacting asperities ($t_c$) and depends on slip rate $v$ in (m s$^{-1}$), material properties (in particular thermal diffusivity $\kappa_{th}$ in (m$^2$ s$^{-1}$)) and asperity radius $r$ in (m).

Strong frictional weakening happens when the temperature rise at the contacting asperity reaches values close to the melting or thermal decomposition temperature of the rock, which can be taken equal to 1000 °C for many rock lithologies[2–4].

We considered a frictional interface where the real contact area ($A_r$) is only a fraction of the nominal contact area ($A$)[29,30]. The major part of the contact is held by asperities that deform mostly plastically and are stressed closely to their yield strength[29,30]. For simplicity, we considered rounded asperities of radius ($r$) and height ($h$). In the presence of fluid pressure, we defined a volume of water ($V_w$) that interacted thermally with the highly stressed asperities such that $V_w$ corresponded roughly to the volume of water displaced by the contact sliding during its lifetime. Such volume of water is in convective contact with the asperity and is defined[7] as $V_w = h.\pi.((2r)^2 - r^2)$ (Fig. 4b).

To compute the temperature elevation per unit surface at a contacting asperity during slip acceleration, we consider that this elevation is due to diffusion of a heat source rate $\tau_c \times v$ where $\tau_c$ is the shear stress at the contact and $v$ is an arbitrarily increasing slip velocity. As argued by Rice[2], the heat input per unit surface over the contact time (and so, until weakening takes place) is directly related to time of contact defined as: $t_c = r/v$. Then we define the flash temperature rise as a heat input term due to shear at the slip rate $v$ and a temperature buffering term[7] due to the volume of water surrounding the asperities $V_w$ as defined in the geometry and shown in Fig. 4b such that: Tflash $= f(\tau_c, v) - g(T, \rho_w(P, T), c_{pw}(P, T))$.

Notice that in this simple model we did not consider the evolution of density or specific heat of solid asperities with temperature (see final equation) and that possible dynamic changes of contact hardness and other material properties due to the flash temperature rise[31] were neglected. Here the fluid pressurized in the fault zone is water. The isobaric evolution of water's specific heat and density with temperature at the experimental pressures were taken from NIST database for thermophysical properties of fluids[24] (based on the IAPWS97 industrial thermodynamic formulation) at different imposed fluid pressures.

The following considerations were used for this model:

Asperities of radius $r$ and height $h$. In a frictional interface, the real contact area of the two surfaces involved is substantially smaller than the apparent contact area[29,30]. Therefore, the load supported by each contacting asperity is considerably higher than the normal stress applied to the apparent surface. In our experiments, microstructural analysis (Fig. 3) showed an initial asperity sizes of ~2–40 μm (Fig. 3a). After deformation, the melted patches in dry and low fluid pressure experiments had maximal sizes of ~20× 20 μm$^2$. Therefore, we defined the maximum asperity size of a radius $r = 20$ μm and an asperity height $h = r = 20$ μm.

Applied forces considered: shear and normal stress. Here peak shear stress considered for all simulations matched the average peak static shear stress found during low and high pore pressure experiments, $\tau_0 = 70$ MPa. The peak friction reached was averaged to $\mu_0 = 0.7$ respecting 'Byerlee's rule'[21]. Therefore, the peak static normal stress considered in this model was: $\sigma_{n0} = \tau_0/\mu_0 = 100$ MPa. If $\alpha = A_r/A$ is the ratio between real contact area and nominal contact area, it writes $\alpha = P_m/\sigma_{n0}$[29]. Where $P_m = 6$ GPa is the estimated penetration hardness of WG taken as a weighted average[32] of hardnesses of the minerals present in the granite. The shear stress held by a single asperity writes then $\tau_c = \alpha.\tau_0 = 4.2$ GPa.

Pure diffusion in the vicinity of the contacts surrounded by pressurized fluid. The question arises whether cooling process by convection of water surrounding the asperities is a purely advective, mixed advective/diffusive or purely diffusive process. Owing to the intense and fast heating developed at the highly stressed contacts, the interacting water volume $V_w$ was considered instantaneously in thermodynamic equilibrium prior to weakening. The temperature of the fluid is therefore that of the contacts. We then calculated the ratio between the time needed for temperature at asperities to equilibrate with water by advection ($t_{adv}$) and the time needed for temperature at asperities to diffuse in water ($t_{heat}$) (Supplementary Fig. 5). In the same water volume, the ratio: $t_{adv}/t_{heat}$ is expressed as the inverse ratio of hydraulic and thermal diffusivities, respectively. $D_{hy} = k(\eta^\star.\beta)^{-1}$ is the hydraulic diffusivity where $k$ is the in plane fault's permeability, $\eta^\star$ is the fluid viscosity and $\beta$ is the storage capacity of the interacting volume. $D_{th} = \lambda^\star(\rho_w^\star.c_{pw}^\star)$

$^{-1}$ is the thermal diffusivity of the fluid volume, where $\lambda^\star$ is thermal conductivity of the fluid, $\rho_w^\star$ is the fluid density and $c_{pw}^\star$ is the fluid specific heat. All values marked with $\star$ are dependent on temperature[25]. The higher the $t_{adv}/t_{heat}$ ratio, the more the cooling process is diffusive. On the other hand, for low values of $t_{adv}/t_{heat}$ (<1), the cooling process should be highly enhanced by fluid circulation in the fault and so it becomes an advective process. These calculations showed that the heating process is purely diffusive in the low pressure case (1 MPa) for fault permeabilities <10$^{-17}$ m$^2$ in the whole temperature range. At high fluid pressure (25 MPa), the process is purely diffusive for permeabilities <10$^{-18}$ m$^2$. At the normal stresses developed in our experiments, in the absence of fault gouge, we estimate that the permeability of the fault was close to that of the surrounding material and so close to values probably inferior to 10$^{-18}$ m$^2$. We conclude that a purely diffusive model represents well the cooling effect of water during FH in the vicinity of asperities. Thus the model assumed that a finite volume of water in the vicinity of the asperity interacted thermally with the latter through heat capacity and latent heat of vaporization (Fig. 4c–e). Calculations presented in Supplementary Fig. 5 are in agreement with the experimental results while fault permeabilities are >10$^{-17}$ m$^2$ for low fluid pressure. In the case of high fluid pressure experiments, the cooling process should be enhanced by advection around the contacts at the temperature of the liquid/supercritical transition for fault permeabilities reaching 10$^{-19}$ m$^2$.

Finally, using the stated parameters and considerations, a heat balance per unit area at the asperity where the heat stored in the asperity ($V_{asp}.\rho.c_p$.Tflash) equals the heat production at the asperity ($\tau_c.v.t_c.A_{asp}$) minus the heat buffer due to the interacting water volume ($V_w.\rho_w(c_{pw}.T + L_w)$) yields:

$$V_{asp}.\rho.c_p.\text{Tflash} = \tau_c.v.t_c.A_c - V_w.\rho_w(c_{pw}.T + L_w) \qquad (4)$$

where $V_{asp} = A_{asp}\sqrt{\kappa_{th}.\pi.t_c}$ is the heated solid volume and $A_{asp} = \pi.r^2$ is the heated area of the asperity.

Therefore, we computed the flash temperature rise at the contacts at equilibrium following:

$$\text{Tflash} = \left(\frac{1}{\rho.c_p\sqrt{\kappa_{th}.\pi}}\right)\left(\tau_c.v\sqrt{t_c} - \frac{V_w.\rho_w(P,T)}{t_c.\pi.r^2}\left(T.c_{pw}(P,T) + L_w(P,T)\right)\sqrt{t_c}\right)$$

$$\qquad (5)$$

where water density ($\rho_w$), specific heat ($c_{pw}$) and latent heat ($L_w$) evolved with pressure and temperature through the thermophysical evolution interpolated from data of NIST [25], shown in Fig. 4c–e. $\kappa_{th}$ is the rock's thermal diffusivity. Parameter values are given in Supplementary Table. 2.

This idealized model accounted for temperature buffering from asperities by the fluid volume in purely diffusive interaction with it. Supplementary Table 2 presents the values used for calculations that resulted in the flash temperatures of Fig. 4a. The following limitations are noticeable: First, this idealized model did not account for reduction of normal stress due to TP of fluid. Instead, we imposed a constant normal stress with increasing slip in order to observe the theoretical flash temperature reached at the asperities. In our experimental conditions, since the laboratory earthquakes nucleated and arrested spontaneously, we expect slip rates to increase during the dynamic stress drop (from $\tau_0$ to $\tau_{min}$), accommodating most of the event slip. Then, a deceleration of the slipping zone is expected to occur during the healing phase (from $\tau_{min}$ to $\tau_f$) and so a very fast reduction of shear heating is expected until temperatures slightly higher than room temperature. Our model accounted only for the first phase, where the fault slips at constant slip rate. During the rupture arrest phase, the reversibility of the vaporization process should account for fast cooling of the melted asperities and significant reduction of the pressurized volume with an increase in normal stress, arresting fault weakening and increasing fault's strength. Second, further considerations of permeability, porosity and other properties of the rock during mechanical changes induced by seismic slip and the rupture passage are not considered in our calculations[31]. Nevertheless, combining the mechanical results recorded dynamically during earthquake rupture, the observed microstructures and our idealized models bring major insights to the interaction between fault fluids and the weakening mechanisms activated thermally during seismic slip. Note that here vaporization of fault water is reflected in the jump in latent heat, which acts as a heat barrier (Fig.4e), therefore the notion of kinetics of this phase transition is not taken into account in this model.

**Bulk fault temperature model.** We considered a one-dimensional macroscopic fault critically stressed at an initial normal stress ($\sigma_n'$) with a friction coefficient of 0.7 (ref. [21]). The fault is sheared at a constant shearing rate $v$ over a thin slip zone of thickness $w_{sz}$[2,15] where the temperature and fluid pressure increase with shear loading. Note that, in the manuscript, the results presented are for a shear zone thickness of 5 μm. For a finite amount of frictional slip ($\delta$), the generated heat ($q = \frac{\tau.\frac{v}{w_{sz}}}{2}$) induces a temperature rise $T$ in the slip zone and then diffuse into the surrounding rock wall. If the bulk fault shear heating phenomenon is drained, we will assume that the fluid pressure in the fault is equal to the initial pore pressure imposed. Conversely, if the conditions are undrained, the generated heat will induce a pore pressure rise ($\Delta$pf). Therefore, the thermophysical properties of fault

water ($\rho_w$, $c_{pw}$, their derivatives and $\eta$) evolve with pressure and temperature[25]. In this model, no chemical reactions are investigated.

In order to quantify the water volume in the fault, a given fault porosity $\varphi$ is imposed and so the specific mass capacity of the bulk fault is a function of porosity such that:[15]

$$\rho_b . c_b = (1 - \varphi)(\rho . c_p) + \varphi(\rho_w . c_{pw}) \qquad (6)$$

In this model, we considered the energy and fluid mass conservation equations (in a similar manner as that of ref. [15]) for the energy and fluid mass conservation in the presence of fluids such that:

$$\frac{\partial T}{\partial t} = \frac{1}{(\rho_b . c_b)} . \mu_0 (\sigma_n - pf)\left(\frac{v}{wsz}\right) + \alpha_{th} . \frac{\partial^2 T}{\partial y^2} \qquad (7)$$

and

$$\frac{\partial pf}{\partial t} = \frac{\lambda_f - \lambda_n}{\beta_f - \beta_n} . \frac{\partial T}{\partial t} + \alpha_{hy} . \frac{\partial^2 pf}{\partial y^2} \qquad (8)$$

where $\alpha_{th} = l_w(\rho_w . c_{pw})^{-1}$ is the thermal diffusivity of the fluid and $l_w$ is the thermal conductivity of the fluid. $\lambda_f$ and $\lambda_n$ are, respectively, the isobaric thermal expansion coefficients of the fluid volume and of the solid pore space. $\beta_f$ and $\beta_n$ are, respectively, the compressibilities of the fluid volume and the solid pore space. And finally, $\alpha_{hy} = k(\eta . \beta)^{-1}$ is the hydraulic diffusivity of the fault where $k$ is the fault's permeability, $\eta$ is the fluid's viscosity and $\beta$ is the compressibility of the fluid volume.

Note that, as discussed by Chen et al.[15], the latent heat (here $L_w$) being constant for pressures lower than that of the supercritical phase transition, it vanishes when deriving the energy conservation equation since $\frac{\partial \rho_w . h_w}{\partial T} = \frac{\partial \rho_w (c_w . T + L_w)}{\partial T} = \rho_w . c_w$, where $h_w$ is the enthalpy of water.

In addition, this model does not consider the kinetics of the vaporization transition but rather an instantaneous phase change when temperature is high enough to overcome this transition.

We solve Eqs. (6)–(8) by an explicit finite difference method. On the spatial boundaries, we impose a no-flow condition. The spatial size of the fault and space step (following $y$ axis) is taken such that pressures and temperatures have reached a constant value far from the boundaries at final time.

When fault slip is a purely drained phenomenon, pore fluid pressure is constant in time on the fault during slip. In this case, our model accounts for the temperature rise at a shear stress across the fault taken equal to the shear strength reached in our experiments ($\tau_0 = 70$ MPa). Then the main assumption is that the shear stress follows the effective stress law such that: $\tau = \mu_0(\sigma_n - pf)$, therefore if the deformation is drained, the stress remains constant. This constant shear stress assumption is not fully realistic but it also gives an upper bound to the temperatures reached during fault slip. The results of this model are presented in Fig. 5a and commented in the main text.

When shear heating due to fault slip is a purely undrained-adiabatic phenomenon, pore fluid pressure evolves with time in the fault during slip, and so the thermophysical properties of pore fluid evolve with rising temperature and pressure. In this case, the shear stress along the fault once again follows the effective stress law described before. Therefore, any increase in pore fluid pressure induces a reduction in effective stress. This case gives us a lower bound for the reached temperature rise if TP is the only weakening mechanism activated during slip. The results of this model are presented in Fig. 5b, c and commented in the main text.

**Extrapolation to upper crustal depths**. In order to extrapolate the model to upper crustal depths, we substituted the experimental stress used in the heat source term of (Eq. 7) for a mean stress taken as the lithostatic overburden gradient of 27 MPa km$^{-1}$ with a fluid pressure gradient of 10 MPa km$^{-1}$, an initial friction of 0.7 and a geothermal gradient of 30 °C km$^{-1}$ (ref. [27]). The results of this extrapolation are presented in Fig. 6 and commented in the main text.

The following limitations are noticeable: The kinetics of the vaporization reaction are not considered in this model. For details on some attempts to constrain such kinetics, refer to ref. [15]. Instead, we have considered that the vaporization reaction is instantaneous and that the latent heat acts as a heat barrier. In the case of the supercritical transition, the terms concerning the kinetics of the reaction vanish[15,16]. In addition, all heterogeneities that exist in fault zones[1] (in terms of thermal, mechanical and hydraulic properties normal and parallel to the fault plane) are neglected in this model.

**Data availability**. Data are available in supplementary materials. Any further information can be requested from the corresponding author.

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

## Acknowledgements

M.A. and M.V. acknowledge the funding support from the Swiss National Science Foundation (SNF), project no. PYAPP2_160588, and the technical staff at LEMR for help with sample preparation. M.V. acknowledges the European Research Council Starting Grant project 757290-BEFINE. A.S. acknowledges the European Research Council, grant no. 681346-REALISM. F.X.P. acknowledges funding from the SNF, project no. PZENP2_173613. M.A., M.V. and A.S. also acknowledge the Germaine de Staël French-Swiss exchange program. The paper was greatly improved thanks to the reviews of D. Garagash and an anonymous reviewer.

## Author contributions

Original idea was from M.V. M.A. and F.X.P. performed the experiments with contributions from A.S. and M.V. M.A. and M.V. performed the microstructural analysis. M. A., M.V. and F.X.P. developed the models for supporting the experimental observations with input of A.S. M.A. and M.V. co-wrote the manuscript with the input of F.X.P. and A.S. All authors participated in the interpretation of the data and the discussion of the manuscript.

## Additional information

**Competing interests:** The authors declare no competing interests.

