## [Peer Review File · Nature Communications]

Reviewers' comments:

Reviewer #1 (Remarks to the Author):

Acosta et al. "Dynamic weakening during earthquakes controlled by fluid thermodynamics" submitted to Nature Comm.

Review by D. Garagash

What are the major claims of the paper?

The paper explores the physics of dynamic fault weakening during earthquakes on laboratory faults. The nature (mechanisms) of this weakening as related to the frictional heating during fast slip have been theoretically suggested before, but the direct evidence and validation of the mechanisms (particularly that of the thermal pressurization of the pore fluid) in the laboratory have been scarce, if any. The main achievements of the paper are along the following lines:

- 1) Providing indirect evidence of the thermal pressurization fault weakening during laboratory earthquakes (inferred from the quantified differences in fault dynamic weakening under dry and saturated conditions).
- 2) Discovering that the extent of the weakening due to the frictional heating during an earthquake can be limited by supercritical transition in the pore fluid, which effectively acts as a heat sink and buffers the further rise of the temperature and of the pore fluid pressure on the fault plane. This limits the extent of the fault weakening by the thermal pressurization of the pore fluid, and, secondly, precludes the flash heating of the fault slip surface asperities.
- 3) Another finding of this study, the inferred vaporization of pore fluid at low ambient pore pressure conditions (1 MPa) leading to the promotion of the flash heating mechanism under wet fault conditions, is interesting, but would appear to be relevant only to very shallow depth. It may however aid the interpretation of the results of a number of other laboratory earthquake studies that have been conducted under similar (dry or low pf) conditions, and their relevance (or lack thereof) to understanding of the natural earthquake phenomena.

Are they (findings of the paper) novel and will they be of interest to others in the community and the wider field?

Yes, the findings are novel and will be of interest to the earthquake physics community, as well as general physics and mechanics as relates to surface friction.

Is the work convincing, and if not, what further evidence would be required to strengthen the conclusions?

The paper conclusion that the supercritical transition of the pore fluid due to frictional heating tends to buffer the temperature rise and therefore to moderate the extent of the TP weakening is supported by experiments and calculations at upto 45 MPa ambient pore pressure corresponding to 4.5 km depth in the crust if the hydrostatic gradient is assumed. The implication of this result to natural earthquakes would be strengthened if the authors extended their analysis (calculations) to pore pressure values more representative of seismogenic depth range with the median ~ 7 km (ambient pressure at or above 70 MPa hydrostatic value). In other words can the authors quantify if the supercritical water transition would have a similarly strong effect on the fault heating at these depths?

On a more subjective note, do you feel that the paper will influence thinking in the field?

I think the paper will have a lasting influence on the field, as it lends credence to and extends the current understanding of the physics of the earthquake source, and the notion of statically-strong while dynamically-weak faults. The idea that thermodynamics of the pore fluid (supercritical

transition during dynamic heating) may play a major role in the earthquake slip development is important to further explore and test.

Comments

I. Can one actually make a quantitative argument based on the observed dynamic strength of the fault, whether or not vaporization of pore fluid leads in the low pf case to enhanced thermal pressurization? It appears not to be that clear of an argument since the minimum dynamic strength is about the same in the dry and low pf cases (Figure 1), thus this would suggest that vaporized fluid offers no additional weakening? Can the vapor overpressure be sustained on the fault plane in view of the drastically increased diffusivity (compared to that of liquid water)?

II.

a) There appear to be a typo in the final equation for the flash heated temperature rise at an asperity (lines 463-464) in the Methods. The heat "sink" (2nd term in the right hand side) term should be inversely proportional to the sqrt of the contact time.

b) It would be helpful if the authors actually provided this equation in the form which can be more easily understood, i.e. in the form of the heat balance (i.e. before solving it for T_{flash}). For example,

the heat stored in the asperity, $(V_{\text{heated_solid}}) \rho c T_{\text{flash}}$,
where $V_{\text{solid_heated}} = A_c h_{\text{heated}}$, and $h_{\text{heated}} = \sqrt{(\pi k t c)}$ is the "heated" height of the asperity with contact surface A_c ,

equal to

the heat production, $\tau v t c A_c$,

minus

the heat buffered by fluid, $V_w \rho_w (c_w T + L_w)$.

c) Acknowledge that in the absence of buffering (i.e. dry case) the first term corresponds to the flash temperature from the exact solution for the 1D-diffusion from slip on a plane (of a contact), as discussed by Rice [JGR 2006] and Proctor et al [JGR 2014], assuming constant values for the contact stress and if the slip velocity.

d) If I understood correctly, the heat sink to the pore fluid in the equation for T_{flash} is modeled assuming the isothermal "bath" conditions in fluid volume V_w exchanging heat with the asperity. Why is that a sound assumption? (Estimates of the thermal diffusivity in the solid asperity and in water seem to suggest that the latter may be much smaller than the former, thus, how likely would isothermal fluid bath conditions may be?)

III. When modeling thermal pressurization to have occurred in the experiments (Methods Section), the use is made of the flash temperature rise (lines 503-504). Can the authors clarify this, as it seems to conflict the meaning of the "flash" temperature rise, understood to be the spatially (and temporally) localized rise at contacting asperities, which constitute only few percent of the fault surface. Thus, the flash temperature rise is expected to be much larger than the actual "macroscopic" temperature rise at the slip surface. And it is the latter (not the former) that should define the thermal pressurization of pore fluid, as in $d(\text{pf}) = \Lambda dT$. (The slip on the plane solution of Rice [2006], provides the full prediction of both macroscopic fault temperature and pore fluid pressure based on the accrued slip = $v t c$)

Minor comments

Line 89: "exponentially" should be changed to "as a power law (of slip rate)"

Reviewer #4 (Remarks to the Author):

This paper describes the results of novel friction experiments on bare rock surfaces under realistic stress conditions, and they found dramatic frictional weakening during dynamic rupture, which are quite different under various fluid pressure conditions. The key observation is that they found the dynamic stress drops are related to the fluid pressures. They claimed the first evidence for thermal pressurization during dynamic rupture and that dynamic weakening during earthquakes is controlled by fluid thermodynamics, especially heat capacity. To confirm this, this manuscript contains some simple calculations on "flash temperature" and "thermal pressurization". The experiments and data processing read great to me (Methods). The mechanical data is sound and convincing (Figures 1 - 2). However, some of their calculations and interpretation do not make sense to me (Figure 4). In other words, all the data are excellent, new, and well-conceived, but, calculation and interpretation needs to be improved (e.g. by rigorous numerical modeling). Moreover, the importance of phase transition is probably limited to relatively shallow depths. Notwithstanding these issues, the paper is of great interests to both experimental and geophysical communities. It would be suitable for publication in Nature Communications. Here are my detailed general and minor comments:

1) Rigorous numerical modeling to validate the importance of fluid thermodynamics on dynamic weakening is lacking. The paper refers to the recent work by Chen et al., (2017, JGR), which investigates the effects of liquid-to-vapor transition of water on seismic slip at relevant conditions. This paper's observation (conclusion) from experiment under low pore pressure (1 MPa) is mostly consistent with the model prediction given by Chen et al.. Regarding the importance of liquid-to-supercritical transition of water, I think the authors missed another (perhaps more important) modeling paper by Urata et al. (2015), also published in JGR. Some of the main conclusions from the high Pf experiment in the paper have been emphasized by Urata et al. (2015). Of course, the present paper has provided unique experimental observations on the importance of fluid thermodynamics on dynamic weakening. The authors have also made significant efforts to relate the distinct observations under different Pf conditions to some water properties (i.e. heat capacity) (see "Flash temperature calculation" and "Thermal pressurization calculation" in the Methods and Supplements). However, both modeling papers aforementioned have considered the full thermodynamic properties of water during seismic slip. I suggest similar modeling work to be done for the present experiments. I understand the present calculation used flash heating as the heat source, which is an entirely new concept and should be different from the modeling of bulk thermal pressurization (cf. Urata et al., 2015). However, the bulk modeling with full thermodynamic properties of water can provide a lower bound but qualitative value on how much weakening is attributed to fluid pressurization (the present calculation on thermal pressurization is too simple). I appreciate adding rigorous modeling work also because it can help validate the hypothesis proposed in the paper that "heat capacity" is the controlling parameter, as is related to my next comment.

2) "Heat capacity" may be not significantly more importance than other properties (e.g. density or its derivatives such as compressibility), and it has been emphasized by previous studies. According to the theory of thermal pressurization, the involved fluid properties for fluid pressurization calculation include viscosity, heat capacity, thermal expansivity, isothermal compressibility (e.g. Rice, 2006). The importance of these parameters have been emphasized in previous numerical work, by performing numerical tests under constant versus state-dependent water properties [Tanikawa, et al., 2009; Chen et al., 2013; Urata et al., 2015]. It is still not conclusive that which parameter controls or contributes more to the dynamic weakening (or temperature buffering). The present paper seems to emphasize the importance of "heat capacity". However, the methodology that the present work used is not significantly better than the previous work (Figure 4). First,

Figure 4 shows the heat capacity have abrupt changes around the critical state. However, it is not clear that density is less important than heat capacity just from the two curves (if you do the derivatives of density to P and T, you will also get very sharp changes in compressibility and expansivity as well). Second, the importance of density and heat capacity beyond the critical state have been emphasized by Chen et al. [2017] (see their Figure 14 and discussions in section 5.3.1). The difference is that Chen et al. [2017] used the terms "density" and "enthalpy" (see also their Figure 2), and emphasized their derivatives. Actually, enthalpy is a higher order expression of thermal energy than temperature, which can be expressed as

$$h = cT + h_0,$$

where h_0 is the reference value at a given condition. Therefore, specific heat capacity (c) is a reflection of enthalpy. Anyway, I agree with the authors that heat capacity is a very potential heat sink, but it is tricky to say that heat capacity is the controlling parameter beyond the liquid-supercritical phase transition.

3. The rapid change of heat capacity (and density) is probably limited to a narrow zone beyond the critical point, say from 22 to 50 MPa (Chen et al., 2017, and your extended-data Figure 3 at $P_f = 45$ MPa). Therefore, its importance on "controlling earthquake dynamic weakening" needs to be clarified.

4. The argument and calculation on "pure TP before vaporization could account for most of the weakening", as given in Methods and Extended-Data Figure 7, is not convincing. As the authors already said in the text, "TP leads to a maximum fluid pressure rise of ~ 23 MPa, therefore the enhanced dynamic stress drop and smaller μ_d observed at LowPf compared to dry conditions are well explained by early TP effect. First, 23 MPa is a maximum, not a minimum, so this argument is not logic. Second, the calculation is not from rigorous coupled modeling using full thermodynamic properties of water, and a random TP coefficient (λ) is used. To some sense, I agree that if the authors simulated this process more rigorously, probably they can still get high pressurization from pure TP. The most important reason I am against from this point is from the procedure used in the experiment, and also from our own experience. Just from the procedure that the authors used to saturate the sample, the sample can still contain some air in it. This is probably not important for calculate temperature rise during dynamic rupture, but it could be crucial for calculating fluid pressurization, since the compressibility of air is much higher than pure water. If you consider a small amount of air in the calculation, the efficiency of TP would be significantly reduced. One test is probably needed to calibrate the compressibility of fluid within the sample (c.f. Violay et al., 2015, EPSL).

5. I enjoy reading your calculation of flash temperature in the presence of water. I have one little concern about the scale of the asperities. You used the maximum asperity size in your calculation (20 micron). In the extended calculation, you vary the size between 10 to 40 microns. However, from your microstructures, the asperities are quite small (Figure 3 and Extended Figure 4), especially for the high-Pf sample (Figure 3d). Let us say an average of less than 5 micron. Can you explain that? If you use large asperity size (20 micron) and assume near-yielding local stress, how many asperities can you get for your slip surface? It is that realistic?

Another concern is the flash temperature calculation for the low-Pf condition. What is the physics behind the calculation "Larger slip velocities (>5 cm/s) allow water to overcome the phase transition temperature"?

Previous TP models have considered phase transition but did not incorporate FH. Your methods and concept here is totally new. My third concern is that your calculations show that heat capacity is important in buffering the flash temperature, but how about the local fluid pressurization? We know it is a coupled process. Presumably, your calculation using flash temperature as heat source will overestimate the local pressurization effect. Moreover, your calculation used a constant TP coefficient of $0.34 \text{ MPa } ^\circ\text{C}^{-1}$, which might be not applicable to your experimental conditions.

My final question is that you can probably calculate local thermal pressurization effects (extended Figure 7b), but how do you determine the dynamic friction and stress drops (Figure 7a and 7c)? There is a scale difference, is not it? Or you assume the local fluid pressure at the asperity scale is

consistent with bulk fluid pressure?

Minor comments (most are just repeating my major comments)

The term "TP" is confusing in the context. In some cases (e.g. Lines 10-13, and 29-31), TP is used as a term excluding phase transition. In some cases, you consider liquid-supercritical transition as a TP process (like Line 22), but in other cases, you consider liquid-vapor transition is not a TP process (Line 110).

Line 10. (i.e. flash heating) better use (e.g. flash heating)

Lines 14-15. You probably can not exclude liquid-to-vapor phase transition. "Fluid pressurization" or "phase transition" might be better. I am not sure.

Lines 20-21. This is not consistent with your argument in the text. You said in the text that pure TP can explain most of the weakening.

Lines 23-26. Very suspicious of the applications. The importance of liquid-supercritical phase transition is probably limited to relatively shallow depths (22 – 50 MPa, < 5 km) (Chen et al., 2017, Extended Figure 3).

Lines 80-81. The asperity size is the maximum of the observation (Figure 3).

Lines 91-98. What is physics behind your statement that "Larger slip velocities allow water to overcome the phase transition temperature". It is not clear in this sentence.

Lines 99-102. Yes, heat capacity is important. Density (thermal expansion) of water could have similar effects on reducing the efficiency of FH and hindering T rise. Or, are you saying that fluid pressurization is not important in the asperity scale?

Lines 138-142. The importance of phase transition cannot be readily applied to greater depths like seismogenic zone.

Method

Lines 390-392: explanation of the equation is lacking

Lines 463 464: derivation of the equation is not straightforward. Readers may want to see the derivation of these equations.

Section "Thermal pressurization of pore fluid"

The fluid pressurization effect is controlled by the thermal pressurization coefficient (λ). It depends on the thermal and transport properties of the rocks as well as on the fluid properties. Using a 0.34 MPa.C-1 may not make sense for your experiments.

Extended Figure 7. Typo: "Pressurization"

Urata, Y., K. Kuge, and Y. Kase (2015), Effect of water phase transition on dynamic ruptures with thermal pressurization: Numerical simulations with changes in physical properties of water, *J. Geophys. Res. Solid Earth*, 120, doi:10.1002/2014JB011370.

I am looking forward to seeing the revised version.

Response to Reviewers' comments:

Please find attached our revised manuscript "Dynamic weakening during earthquakes controlled by fluid thermodynamics". The comments of the reviewers were very constructive. They helped clarifying the content of the manuscript. We performed an in-depth revision of our manuscript to address all comments. The point-by-point answer is reported below, with the reviewers' comments in black font and our replies in red. Replies are numbered in order to track them in an easy manner.

The main changes that have been made are the following:

- i. We, performed new calculations of flash temperature at the asperity level. Results are presented in Figure 4 of the revised manuscript. Several pore pressures up to 70 MPa (~7 km depth) were investigated. The main observation is that under flash heating theory, water's density and heat capacity act as a heat buffer with a maximum efficiency around water's liquid-supercritical phase transition (see Supplementary Figure 7). Such calculations are described and discussed in the revised manuscript in lines 76 – 110 and the implications at higher depths in lines 142-157.*
- ii. We developed a new numerical model of shear heating of the bulk fault through a finite difference numerical model (Methods) in both drained (Figure 5.a) and undrained conditions (Figure 5.b, c). This model takes into account the effect of pore fluid increases with increasing temperature, which is in turn used to compute stress evolution following the effective pressure law. Then, we extrapolated this bulk fault temperature model to upper crustal conditions down to 7 km depth. (Figure 6). The main results observed from these new computations are that, as described in lines 122 to 167 (i) under drained conditions (Figure 6a), the water heat buffer effect is still observed (ii) under undrained conditions, the temperature rises are governed by a competition between the heat source term which becomes very large with depth and the water heat buffer. (Figure 6b). At depths >7km, the heat buffer effect fades and gives place to thermal decomposition temperatures in the overall fault. (iii) the friction drop is of the same magnitude at all initial fluid pressures because, for small slip, thermal pressurization of pore fluid is controlled by a small weakening strain scale as discussed by Brantut and Platt, 2017¹².*
- iii. We added figure 7 in Supplementary so that the interested reader can see how the thermophysical properties of water evolve with pressure and temperature.*

We hope that this improved version will provide a more complete picture of our experimental results, the theoretical and numerical models and their implications for earthquake mechanics.

With best regards,

Mateo Acosta (on behalf of all the co-authors)

Reviewer #1

What are the major claims of the paper?

The paper explores the physics of dynamic fault weakening during earthquakes on laboratory faults. The nature (mechanisms) of this weakening as related to the frictional heating during fast slip have been theoretically suggested before, but the direct evidence and validation of the mechanisms (particularly that of the thermal pressurization of the pore fluid) in the laboratory have been scarce, if any. The main achievements of the paper are along the following lines:

1) Providing indirect evidence of the thermal pressurization fault weakening during laboratory earthquakes (inferred from the quantified differences in fault dynamic weakening under dry and saturated conditions).

2) Discovering that the extent of the weakening due to the frictional heating during an earthquake can be limited by supercritical transition in the pore fluid, which effectively acts as a heat sink and buffers the further rise of the temperature and of the pore fluid pressure on the fault plane. This limits the extent of the fault weakening by the thermal pressurization of the pore fluid, and, secondly, precludes the flash heating of the fault slip surface asperities.

3) Another finding of this study, the inferred vaporization of pore fluid at low ambient pore pressure conditions (1 MPa) leading to the promotion of the flash heating mechanism under wet fault conditions, is interesting, but would appear to be relevant only to very shallow depth. It may however aid the interpretation of the results of a number of other laboratory earthquake studies that have been conducted under similar (dry or low pf) conditions, and their relevance (or lack thereof) to understanding of the natural earthquake phenomena.

Are they (findings of the paper) novel and will they be of interest to others in the community and the wider field?

Yes, the findings are novel and will be of interest to the earthquake physics community, as well as general physics and mechanics as relates to surface friction.

#1- We would like to thank the reviewer for his thorough revision, and for his positive comments.

Is the work convincing, and if not, what further evidence would be required to strengthen the conclusions?

The paper conclusion that the supercritical transition of the pore fluid due to frictional heating tends to buffer the temperature rise and therefore to moderate the extent of the TP weakening is supported by experiments and calculations at up to 45 MPa ambient pore pressure corresponding to 4.5 km depth in the crust if the hydrostatic gradient is assumed. The implication of this result to natural earthquakes would be strengthened if the authors extended their analysis (calculations) to pore pressure values more representative of seismogenic depth range with the median ~ 7km (ambient pressure at or above 70 MPa hydrostatic value). In other words can the authors quantify if the supercritical water transition would have a similarly strong effect on the fault heating at these depths?

#2- We would like to thank the reviewer for this comment. We have modified the manuscript, the methods and the supplementary figures in order to evaluate the influence of depth in the heat-buffer effect of the liquid-supercritical transition. To this end, we have thoroughly reviewed the manuscript as discussed in the description of modifications above. The depth dependence of the heat buffer effect are now discussed in lines 142 – 157 of the revised manuscript as follows:

“Similar stress evolutions observed in experiments conducted at other effective stresses and at $pf=45$ MPa (Extended-Data Fig. 3) suggest that the observed heat buffer operates even at higher fluid pressures, where the liquid-supercritical transition is smoother²⁴ (e.g. Fig. 4c, d and Supplementary Fig. 7). To further study the depth dependence of this heat buffer effect, we computed again the temperature rises (in both drained (Fig. 6a) and undrained conditions (Fig 6b, c)) and effective stress variations due to TP with a depth extrapolation for a mean stress equal to the lithostatic overburden gradient of 27 MPa/km, a hydrostatic fluid pressure rise of 10 MPa/km and an initial friction of 0.7 (e.g. Methods). When considering a depth dependence of shear stress, we observe that a heat buffer can operate for fluid pressures up to 45 MPa in both drained and undrained cases but its effectiveness is strongly reduced when fluid pressures reach 70 MPa at 7 km depth. This, because higher background stress and a smoother supercritical transition allow to overcome the transition temperature for smaller slips when sliding at seismic slip rates ($\sim 1\text{m/s}$), consistently with previous studies on the depth dependence of weakening mechanisms¹². Nevertheless, the dynamic friction values predicted by TP theory are similar at all depths for a given final slip, likely because at higher depths ($> 7\text{km}$), the background driving stress has a stronger effect than the pore fluid pressure rise on TP at small slip¹².”

On a more subjective note, do you feel that the paper will influence thinking in the field?

I think the paper will have a lasting influence on the field, as it lends credence to and extends the current understanding of the physics of the earthquake source, and the notion of statically-strong while dynamically-weak faults. The idea that thermodynamics of the pore fluid (supercritical transition during dynamic heating) may play a major role in the earthquake slip development is important to further explore and test.

#3- We would like to thank the reviewer for this encouraging comment, and hope that these novel findings will help improving the understanding of how faults weaken and the controls on earthquake propagation.

Comments

I. Can one actually make a quantitative argument based on the observed dynamic strength of the fault, whether or not vaporization of pore fluid leads in the low pf case to enhanced thermal pressurization? It appears not to be that clear of an argument since the minimum dynamic strength is about the same in the dry and low pf cases (Figure 1), thus this would suggest that vaporized fluid offers no additional weakening? Can the vapor overpressure be sustained on the fault plane in view of the drastically increased diffusivity (compared to that of liquid water)?

#4- We agree with the reviewer. We have strongly moderated our statements regarding the additional weakening observed in Low Pf conditions in lines 111 -121. Nevertheless, recent high velocity friction experiments have consistently shown additional weakening due to coexisting melting and thermal pressurization mechanisms (Violay et al. 2015⁸ and Chen et al. 2017a¹⁴). It remains difficult to evaluate if vapor overpressure can be sustained in the fault during seismic slip. However, if it is true in high velocity friction experiments, it should remain true in our experiments due to the extremely short duration of our events compared to possible fluid diffusion (slip duration of $\sim 20 \mu\text{s}$). The modifications in lines 111 -121 are as follows:

“The liquid-vapour transition has been thought to have strong thermal effects on faulting, inhibiting temperature rise while promoting thermal expansion of fluids and their pressurization during co-seismic slip^{14,15}. In high velocity friction experiments^{8,14}, water vaporisation enhanced the friction drop from 0.1, which is comparable to the difference observed between the dynamic friction recorded during Low_{pf} and dry conditions (figure 2b). Such effect could also be due to a reduction of melt viscosity through hydration in the presence of fluids. However, rotary shear experiments have demonstrated that the chemical compositions of melts developed after long slip times (>10 s) under vacuum, room humidity, and fluid-saturated conditions were identical⁷, discarding the possibility of melt-hydration in our experiments (here, the total slip time was <30 μs). TP could then be a candidate to explain the slightly lower dynamic friction values observed at Low_{pf} while FH remains the dominant weakening mechanism.

a) There appear to be a typo in the final equation for the flash heated temperature rise at an asperity (lines 463-464) in the Methods. The heat “sink” (2nd term in the right hand side) term should be inversely proportional to the sqrt of the contact time.

#5- Yes, in fact this was a typo. It has been corrected in line 534 (eq. 5) as:

$$T_{flash} = \left(\frac{1}{\rho_w \cdot c_p \cdot \sqrt{\kappa \cdot \pi}} \right) \left(\tau \cdot v \cdot \sqrt{t_c} - \frac{V_w \cdot \rho_w(P, T)}{t_c \cdot \pi \cdot r^2} \cdot (T \cdot c_{pw}(P, T) + L_w(P, T)) \cdot \sqrt{t_c} \right)$$

b) It would be helpful if the authors actually provided this equation in the form which can be more easily understood, i.e. in the form of the heat balance (i.e. before solving it for T_{flash}). For example,

the heat stored in the asperity, (V_{heated_solid}) ρ_w c_p T_{flash},
 where V_{solid_heated} = A_c h_{heated}, and h_{heated} = sqrt(κ t_c) is the “heated” height of the asperity with contact surface A_c,

equal to

the heat production, τ v t_c A_c,

minus

the heat buffered by fluid, V_w ρ_w (c_{pw} T + L_w).

#6- Yes, we agree with the reviewer. The heat balance and derivation has been added in lines 519-535 before solving for T_{flash} (eq. 4 and 5) as follows:

“Final equation

Finally, using the stated parameters, a heat balance per unit area at the asperity where the heat stored in the asperity (V_{asp} · ρ_w · c_p · T_{flash}) equals the heat production at the asperity (τ · v · t_c · A_{asp}) minus the heat buffer due to the interacting water volume (V_w · ρ_w · (c_{pw} · T + L_w)) yields:

$$V_{asp} \cdot \rho \cdot c_p \cdot T_{flash} = \tau \cdot v \cdot t_c \cdot A_c - V_w \cdot \rho_w \cdot (c_{pw} \cdot T + L_w) \quad (1)$$

Where, $V_{asp} = A_{asp} \cdot \sqrt{\kappa * \pi * t_c}$ is the heated solid volume, and $A_{asp} = \pi * r^2$ is the heated area of the asperity.

Therefore, we computed the flash temperature rise at the contacts at equilibrium following:

$$T_{flash} = \left(\frac{1}{\rho \cdot c_p \cdot \sqrt{\kappa \cdot \pi}} \right) \left(\tau c \cdot v \cdot \sqrt{t_c} - \frac{V_w \cdot \rho_w(P, T)}{t_c \cdot \pi \cdot r^2} \cdot (T \cdot c_{pw}(P, T) + L_w(P, T)) \cdot \sqrt{t_c} \right) \quad (2)$$

c) Acknowledge that in the absence of buffering (i.e. dry case) the first term corresponds to the flash temperature from the exact solution for the 1D-diffusion from slip on a plane (of a contact), as discussed by Rice [JGR 2006] and Proctor et al [JGR 2014], assuming constant values for the contact stress and if the slip velocity.

#7- Yes, we agree with the reviewer, we have added the following sentence (lines 89 - 91):

“Under dry conditions, when no buffering takes place, the flash temperature becomes the exact solution for the one dimensional heat diffusion problem for slip on a plane^{2, 10, 25} at the asperity scale.”

d) If I understood correctly, the heat sink to the pore fluid in the equation for T_{flash} is modeled assuming the isothermal “bath” conditions in fluid volume V_w exchanging heat with the asperity. Why is that a sound assumption? (Estimates of the thermal diffusivity in the solid asperity and in water seem to suggest that the latter may be much smaller than the former, thus, how likely would isothermal fluid bath conditions may be?)

#8- The diffusion length in the solid asperity is given by $\left(\sqrt{\frac{\pi \kappa a}{v}} \right)$. The common flash heating velocities are ~ 0.1 m/s. At such velocities, the diffusion length is ~42 μm . This diffusion length is of about the size of the asperity (~20 μm). In addition, when solid-solid asperities start to slip, they become immediately immersed in the fluid therefore tending towards a fast temperature equilibrium and allowing for isothermal bath conditions. We therefore assumed that the isothermal bath conditions remain valid.

A sentence has been added lines 80-85:

“The main hypothesis of the model is that the fluid volume surrounding the asperities is at thermal equilibrium with the asperity, assumption that should remain valid during frictional slip since, (1) the thermal diffusion length (~42 μm) is close to the asperity size at FH velocity. and (2) when the solid-solid contact starts slipping, a liquid-solid contact forms immediately, allowing for extremely fast temperature equilibrium between the asperity and the surrounding fluid.”

III. When modeling thermal pressurization to have occurred in the experiments (Methods Section), the use is made of the flash temperature rise (lines 503-504). Can the authors clarify this, as it seems to conflict the meaning of the “flash” temperature rise, understood to be the spatially (and temporally) localized rise at contacting asperities, which constitute only few percent of the fault surface. Thus, the flash temperature rise is expected to be much larger than the actual “macroscopic” temperature rise at the slip surface. And it is the latter

(not the former) that should define the thermal pressurization of pore fluid, as in $d(pf) = \Lambda dT$. (The slip on the plane solution of Rice [2006], provides the full prediction of both macroscopic fault temperature and pore fluid pressure based on the accrued slip = $v t_c$)

#9- We agree with the reviewer, that an oversimplification of the problem had been made.

Please refer to modification “ii” in the description of this letter.

We have performed a new numerical model accounting for shear heating of the bulk fault in a 1D finite difference model based on the accrued slip = $v \tau_{U_{bulk}}$ (Methods). In both drained (Figure 5a) and undrained conditions (Figure 5b, c). In particular, under undrained conditions, stress evolution follows the effective pressure law and assumes thermal pressurization equations (Figure 5.b, c). The observations and remarks on these calculations are described lines 122- 141:

“While FH explains the dynamic strength drop observed in dry and Low_{Pf} conditions, it does not explain the small friction drops observed at $High_{Pf}$ conditions. In order to study the small stress drops and slips found at $High_{Pf}$, we computed the temperature rises on a bulk planar fault in both drained (Fig. 5a) and undrained (Fig. 5b,c) conditions through a finite difference numerical model (e.g. Methods). In such model, the full thermodynamic evolution of fluid properties with pressure and temperature are considered^{15, 16, 24}. Under drained conditions, we observe that the reached temperatures (which are a maximum estimation of the possible temperature in the fault since the shear stress for heat generation is taken as the static fault’s shear strength of our experiments) are not high enough to reach a flash heating temperature even for twice the maximum slip observed in the experiments ($\sim 250 \mu m$) (Fig. 5a). Nevertheless, under our experimental conditions, the water heat buffer effect due to the liquid-supercritical transition is still observed for initial pore pressures higher than 22 MPa. Note that strong temperature rises on the fault due to water vaporization take place in a similar manner than for the flash temperature computations, confirming our calculations at the asperity scale. Under undrained conditions (Fig. 5b, c), since the fault’s stress obeys the effective pressure law, we observe an initial fast decay in friction due to TP. The decay then stabilizes leading to friction drops of ~ 0.1 for slips of ~ 20 to $150 \mu m$ in all fluid pressure conditions. Such friction drop values are remarkably consistent with the friction drops observed in $High_{Pf}$ experiments (Fig. 2b). Therefore, at $High_{Pf}$, TP might well be the dominant weakening mechanism in our experiments, as supported by our microstructural analysis.”

Minor comments

Line 89: “exponentially” should be changed to “as a power law (of slip rate)”

#10- Thank you for the observation, the figure has been up-dated and we decided to present the temperature evolution as a function of increasing slip at different constant slip rates. The mistake has been corrected in line 91:

“There, temperature rises as a power law of slip (see Extended-Data Fig. 6 for other asperity sizes).”

Reviewer #4 (Remarks to the Author):

This paper describes the results of novel friction experiments on bare rock surfaces under realistic stress conditions, and they found dramatic frictional weakening during dynamic rupture, which are quite different under various fluid pressure conditions. The key observation is that they found the dynamic stress drops are related to the fluid pressures. They claimed the first evidence for thermal pressurization during dynamic rupture and that dynamic weakening during earthquakes is controlled by fluid thermodynamics, especially heat capacity. To confirm this, this manuscript contains some simple calculations on “flash temperature” and “thermal pressurization”.

The experiments and data processing read great to me (Methods). The mechanical data is sound and convincing (Figures 1 - 2).

However, some of their calculations and interpretation do not make sense to me (Figure 4).

In other words, all the data are excellent, new, and well-conceived, but, calculation and interpretation needs to be improved (e.g. by rigorous numerical modeling).

Moreover, the importance of phase transition is probably limited to relatively shallow depths.

#11- We would like to thank the reviewer for his in depth revision. We believe, we have greatly improved the manuscript, the methods and the supplementary figures by: (i) adding a fully coupled numerical model for shear heating of the bulk fault under both drained and undrained conditions (Figures 5 and 6). The coupled model includes therefore thermal pressurization with full evolution of fluid thermodynamic properties and (ii) studying the water heat buffer effect due to water's phase transitions. (iii) We also include a new analysis on the validity of this model at larger pressures, and extrapolated our results for larger depth.

We wish to underline that the main objective of this paper is to present the experimental results supported by weakening theories and microstructural analysis. Therefore, we believe a full parametrization of a numerical model as proposed by the reviewer might blur the main message of the paper, the experimental observations and difficult its readability.

Notwithstanding these issues, the paper is of great interests to both experimental and geophysical communities. It would be suitable for publication in Nature Communications.

#12- We thank the reviewer for this positive comment.

Here are my detailed general and minor comments:

1) Rigorous numerical modeling to validate the importance of fluid thermodynamics on dynamic weakening is lacking.

#13- We thank the reviewer for his comment. We believe we have answered to the reviewer's comment in reply **#11** and in the introduction of this letter.

The paper refers to the recent work by Chen et al., (2017, JGR), which investigates the effects of liquid-to-vapor transition of water on seismic slip at relevant conditions. This paper's observation (conclusion) from experiment under low pore pressure (1 MPa) is mostly consistent with the model prediction given by Chen et al.

Regarding the importance of liquid-to-supercritical transition of water, I think the authors missed another (perhaps more important) modeling paper by Urata et al. (2015), also published in JGR.

#14- There is a fundamental difference between the prediction of the model by Chen et al. and our experiments because in our experiments, we have found strong evidence that flash heating is the driving mechanism for the large weakening at low pore fluid pressure.

Regarding the modelling paper by Urata et al.¹⁶ Perhaps the reviewer missed reference 24 of the initially submitted manuscript, in fact, the paper by Urata et al (now ref. 16), states that *'the phase transition of pore water has little effect on dynamic ruptures. Fault-normal variations in fluid density and viscosity may play a more significant role.'* Which is consistent with our results in the case that flash heating is not considered.

Some of the main conclusions from the high Pf experiment in the paper have been emphasized by Urata et al. (2015). Of course, the present paper has provided unique experimental observations on the importance of fluid thermodynamics on dynamic weakening. The authors have also made significant efforts to relate the distinct observations under different Pf conditions to some water properties (i.e. heat capacity) (see "Flash temperature calculation" and "Thermal pressurization calculation" in the Methods and Supplements). However, both modeling papers aforementioned have considered the full thermodynamic properties of water during seismic slip. I suggest similar modeling work to be done for the present experiments. I understand the present calculation used flash heating as the heat source, which is an entirely new concept and should be different from the modeling of bulk thermal pressurization (cf. Urata et al., 2015). However, the bulk modeling with full thermodynamic properties of water can provide a lower bound but qualitative value on how much weakening is attributed to fluid pressurization (the present calculation on thermal pressurization is too simple). I appreciate adding rigorous modeling work also because it can help validate the hypothesis proposed in the paper that "heat capacity" is the controlling parameter, as is related to my next comment.

#15- We agree with the reviewer, that an oversimplification of the problem had been made in the initial manuscript. We therefore have performed a numerical modelling work accounting for shear heating of the bulk fault. Please refer to replies **#9**, **#11** and to the description of modifications **i)**, **ii)** in the introduction to this letter.

2) "Heat capacity" may be not significantly more importance than other properties (e.g. density or its derivatives such as compressibility), and it has been emphasized by previous studies. According to the theory of thermal pressurization, the involved fluid properties for fluid pressurization calculation include viscosity, heat capacity, thermal expansivity, isothermal compressibility (e.g. Rice, 2006). The importance of these parameters have been emphasized in previous numerical work, by performing numerical tests under constant versus state-dependent water properties [Tanikawa, et al., 2009; Chen et al., 2013; Urata et al., 2015]. It is still not conclusive that which parameter controls or contributes more to the dynamic weakening (or temperature buffering). The present paper seems to emphasize the importance of "heat capacity". However, the methodology that the present work used is not significantly better than the previous work (Figure 4). However, it is not clear that density is less important than heat capacity just from the two curves (if you do the derivatives of density to P and T, you will also get very sharp changes in compressibility and expansivity as well). Second, the importance of density and heat capacity beyond the critical state have been emphasized by Chen et al. [2017] (see their Figure 14 and discussions in section 5.3.1). The difference is that Chen et al. [2017] used the terms "density" and "enthalpy" (see also their Figure 2), and emphasized their derivatives. Actually, enthalpy is a higher order expression of thermal energy than temperature, which can be expressed as $h = cT + h_0$, where h_0 is the reference value at a given condition. Therefore, specific heat capacity (c) is a reflection of enthalpy. Anyway, I agree with the authors that heat capacity is a very potential heat sink, but it is tricky to say that heat capacity is the controlling parameter beyond the liquid-supercritical phase transition.

#16- The review is right. We performed rigorous coupled numerical modelling for thermal pressurization of the bulk fault, including the main derivatives of density in a similar manner as Chen et al¹⁵ and Urata et al¹⁶. Under thermal pressurization theory, compressibility and expansivity of the rock/fluid system play a major role. However, under flash heating theory

(which is the main mechanism activated during the LowPf test), the parameters that come into play are heat capacity and density. In particular their coupled evolution, because they are the main parameters regulating shear heating (in the case of solid) or temperature buffering (in the case of fluid). The importance of the coupled evolution of water's density and specific heat has been discussed in lines 89 – 110. Once again, a full parametric analysis similar to that of Chen et al²³ or Urata et al²⁵ has not been performed in the present paper because we have access to most of the dominant parameters from our experimental data. In addition, the modelling work was conducted as support for the experimental observations. We believe that adding more modelling results will blur the message of the paper, which was written to present new experimental observations.

The discussion on the importance of the coupled evolution of water's density and specific heat has been discussed in lines 89 – 110 as follows:

“Under dry conditions, when no buffering takes place, the flash temperature becomes the exact solution for the one dimensional heat diffusion problem for slip on a plane^{2, 10, 25} at the asperity scale. There, temperature rises as a power law of slip (see Extended-Data Fig. 6 for other asperity sizes). The expected FH temperature (approximately 1000 °C²⁻⁵) was reached for slip rates >10 cm.s⁻¹ during the asperity lifetime, as predicted by FH theories and experiments^{4,6,7}. At those velocities, in the LowPf case, water-buffered temperatures were observed in the first half of the contact lifetime, and so, flash temperatures remained lower than 179 °C, i.e., while water stayed in a liquid state. Longer slip at such seismic slip velocities allowed water to overcome the liquid-vapour phase transition temperature during t_c , inducing a strong drop in ρ_w and c_{pw} (roughly falling to 0.5% and 50% of their room temperature values respectively; Fig. 4c,d), thereby enhancing shear heating at contacts and leading to flash temperatures similar to those estimated for dry conditions. Conversely, at fluid pressures ranging from 25 to 70 MPa, temperature rise is strongly buffered by water cooling during t_c due to the liquid-supercritical transition. This phase change requires larger amounts of energy because the heat capacity of water increases by 1400% during the transition at $p_f=25$ MPa (Fig. 4c) while the drop in density is smoother than in the case of vaporization²⁴. There, water becomes an extremely efficient energy buffer, reducing the efficiency of FH and hindering rises in temperature higher than that of the liquid-supercritical phase transition (~373 °C at $p_f=25$ MPa, Supplementary Figure 7) at asperity contacts during their lifetime, even for slip rates of 1 m.s⁻¹ (the admitted slip rate during regular earthquakes^{4,6} is ~1 m.s⁻¹). This major heat sink and the resulting temperature buffer explain (i) the reduced dynamic weakening observed at HighPf, and (ii) the absence of frictional melt on the fault surfaces.”

3. The rapid change of heat capacity (and density) is probably limited to a narrow zone beyond the critical point, say from 22 to 50 MPa (Chen et al., 2017, and your extended-data Figure 3 at Pf = 45 MPa). Therefore, its importance on “controlling earthquake dynamic weakening” needs to be clarified.

#17- We agree and thank the reviewer for this comment. We have modified the manuscript, the methods and the supplementary figures in order to evaluate the influence of depth on the water heat-buffer effect of the liquid-supercritical transition. Please Refer to the introduction of this letter and replies **#9**, **#11** for details.

4. The argument and calculation on “pure TP before vaporization could account for most of the weakening”, as given in Methods and Extended-Data Figure 7, is not convincing. As the authors already said in the text, “TP leads to a maximum fluid pressure rise of ~ 23 MPa, therefore the enhanced dynamic stress drop and smaller μ_d observed at LowPf

compared to dry conditions are well explained by early TP effect. First, 23 MPa is a maximum, not a minimum, so this argument is not logic. Second, the calculation is not from rigorous coupled modeling using full thermodynamic properties of water, and a random TP coefficient (Lamda) is used. To some sense, I agree that if the authors simulated this process more rigorously, probably they can still get high pressurization from pure TP.

#18- We have now performed rigorous numerical modelling for thermal pressurization of the bulk fault, including the main derivatives of density (compressibility and expansivity), as well as viscosity in the model. (Eq. 7 and 8; figures 5 and 6). In addition, the arbitrary TP coefficient is not in use anymore since we are now using the full evolution of water's compressibility and expansivity in the new computations for the bulk fault temperature as described in Methods and in previous replies (**#9; #16; #17**; description of modifications (i, ii)). We have also added the evolution of density, specific heat and viscosity with pressure and temperature, in Supplementary Figure 7.

The most important reason I am against from this point is from the procedure used in the experiment, and also from our own experience. Just from the procedure that the authors used to saturate the sample, the sample can still contain some air in it. This is probably not important for calculate temperature rise during dynamic rupture, but it could be crucial for calculating fluid pressurization, since the compressibility of air is much higher than pure water. If you consider a small amount of air in the calculation, the efficiency of TP would be significantly reduced. One test is probably needed to calibrate the compressibility of fluid within the sample (c.f. Violay et al., 2015, EPSL).

#19- We disagree with the reviewer since at closing the outlet valve we are sure that no air is left in the sample because we are flushing from the bottom-up for long periods of time (~2-4 hours), so all air in the sample is replaced by water. In addition, as the fault is a preferential fluid path, pressurized water should be the main fluid in the fault during the experiment.

5. I enjoy reading your calculation of flash temperature in the presence of water. I have one little concern about the scale of the asperities. You used the maximum asperity size in your calculation (20 micron). In the extended calculation, you vary the size between 10 to 40 microns. However, from your microstructures, the asperities are quite small (Figure 3 and Extended Figure 4), especially for the high-Pf sample (Figure 3d). Let us say an average of less than 5 micron. Can you explain that? If you use large asperity size (20 micron) and assume near-yielding local stress, how many asperities can you get for your slip surface? It is that realistic?

#20- Thank you for your comment. It is very important not to confuse the terms 'Asperity' and 'Debris' in fact, the asperities (or contacts) which hold the fault together and are sized ~20 to 40 μm , might breakdown generating **debris (of sizes <1 to 5 μm)** Therefore, the sizes of *asperities* we used are based on the observation of the melted patches from Figure 3 which showed melted **asperities of sizes ~20 to 40 μm** . This is in agreement with values used in flash heating theories.

Another concern is the flash temperature calculation for the low-Pf condition. What is the physics behind the calculation "Larger slip velocities (>5 cm/s) allow water to overcome the phase transition temperature"?

#21- Thanks to the pertinent comment of the reviewer, this sentence has been changed in the revised manuscript on lines 96 - 100. To avoid any confusion, we are now presenting in Figure 4 the temperature evolution as a function of increasing slip for different constant slip rates as follows:

“Longer slip at such seismic slip velocities allowed water to overcome the liquid-vapour phase transition temperature during t_c , inducing a strong drop in ρ_w and c_{pw} (roughly falling to 0.5% and 50% of their room temperature values respectively; Fig. 4c,d), thereby enhancing shear heating at contacts and leading to flash temperatures similar to those estimated for dry conditions.”

Previous TP models have considered phase transition but did not incorporate FH. Your methods and concept here is totally new. My third concern is that your calculations show that heat capacity is important in buffering the flash temperature, but how about the local fluid pressurization? We know it is a coupled process.

Presumably, your calculation using flash temperature as heat source will overestimate the local pressurization effect. Moreover, your calculation used a constant TP coefficient of 0.34 MPa C⁻¹, which might be not applicable to your experimental conditions.

#22- Yes, this is true, in fact, since the flash temperature is much higher than the bulk temperature, in the previous oversimplified model we overestimated the bulk pressurization effect. Reason why we now performed a rigorous numerical model coupling bulk shear heating and bulk fault pressurization considering the full evolution of fluid compressibility and expansivity. For more details, please refer to the description of this letter (points i and ii.) and to replies **#9** and **#11**

My final question is that you can probably calculate local thermal pressurization effects (extended Figure 7b), but how do you determine the dynamic friction and stress drops (Figure 7a and 7c)? There is a scale difference, is not it? Or you assume the local fluid pressure at the asperity scale is consistent with bulk fluid pressure?

#23- We agree with the reviewer that the previous model was oversimplified and therefore nor rigorous enough. We have now performed full numerical models accounting for the bulk fault heating and then, we estimated the shear stress drop from a direct effective pressure law in order to estimate the friction drop (Figures 5 and 6).

Minor comments (most are just repeating my major comments)

The term “TP” is confusing in the context. In some cases (e.g. Lines 10-13, and 29-31), TP is used as a term excluding phase transition. In some cases, you consider liquid-supercritical transition as a TP process (like Line 22), but in other cases, you consider liquid-vapor transition is not a TP process (Line 110).

#24- Thank you, this has been corrected in the revised manuscript.

Line 10. (i.e. flash heating) better use (e.g. flash heating)

#25- This has been corrected in the revised manuscript.

Lines 14-15. You probably can not exclude liquid-to-vapor phase transition. “Fluid pressurization” or “phase transition” might be better. I am not sure.

#26- This has been corrected in the revised manuscript.

Lines 20-21. This is not consistent with your argument in the text. You said in the text that pure TP can explain most of the weakening.

#27- The manuscript has been changed and improved as described in the introduction to this letter.

Lines 23-26. Very suspicious of the applications. The importance of liquid-supercritical phase transition is probably limited to relatively shallow depths (22 – 50 MPa, < 5 km) (Chen et al., 2017, Extended Figure 3).

#28- A sentence (line 19-23) has been added in order to moderate the applications of this study:

“ Our results are supported by flash weakening theories modified for pressurized fluids and by numerical modelling of thermal pressurization. We show that water phase transitions control earthquake rupture dynamics by buffering frictional heat. Effect having a maximum efficiency at mid crustal depths (~2 to 5 km), where many anthropogenic earthquakes nucleate.”

In addition, the depth dependence of the heat buffer mechanisms has been assessed and discussed as described in replies **#9** and **#11**.

Lines 80-81. The asperity size is the maximum of the observation (Figure 3).

#29- Please refer to reply **#20**.

Lines 91-98. What is physics behind your statement that “Larger slip velocities allow water to overcome the phase transition temperature”. It is not clear in this sentence.

#30- Thanks to the pertinent comment of the reviewer, this sentence has been changed in the revised manuscript because we are presenting now the calculation in figure 4 since we decided to present the temperature evolution as a function of increasing slip for different constant slip rates improving the readability of the manuscript. In the original manuscript the sentence meant that at those slip velocities, enough heat is generated during contact lifetimes in order to overcome the transition. For details please refer to reply **#21**.

Lines 99-102. Yes, heat capacity is important. Density (thermal expansion) of water could have similar effects on reducing the efficiency of FH and hindering T rise. Or, are you saying that fluid pressurization is not important in the asperity scale?

#31 - Please refer to reply **#16**.

Lines 138-142. The importance of phase transition cannot be readily applied to greater depths like seismogenic zone.

#32 - Please refer to the main modifications of the manuscript “**ii**” referring to the depth extrapolation of the calculations. In addition replies to this have been addressed in comments **#2, #11, and #28**.

Method

Lines 390-392: explanation of the equation is lacking

#33- we agree with the reviewer, the heat balance and derivation have been added before solving for T_{flash} (eq. 4 and 5). Please refer to Reply **#6**.

Lines 463 464: derivation of the equation is not straightforward. Readers may want to see the derivation of these equations.

#34- we agree with the reviewer, the heat balance and derivation have been added before solving for Tflash (eq. 4 and 5). Please refer to Reply **#6** and **#34**.

Section "Thermal pressurization of pore fluid"

The fluid pressurization effect is controlled by the thermal pressurization coefficient (λ). It depends on the thermal and transport properties of the rocks as well as on the fluid properties. Using a 0.34 MPa.C-1 may not make sense for your experiments.

#35- We agree with the reviewer that the previous model was oversimplified and therefore not rigorous enough. We have now performed full numerical models accounting for the bulk fault heating. As described in the previous answers.

Extended Figure 7. Typo: "Pressurization"

Urata, Y., K. Kuge, and Y. Kase (2015), Effect of water phase transition on dynamic ruptures with thermal pressurization: Numerical simulations with changes in physical properties of water, J. Geophys. Res. Solid Earth, 120, doi:10.1002/2014JB011370.

#36- Yes thank you, this reference was part of the initial manuscript (reference 24, now has become reference 16), perhaps the reviewer missed the reference.

I am looking forward to seeing the revised version.

#37- Thank you for the pertinent comments, we have greatly improved the manuscript, the models, figures, methods and supplementary information. We hope it meets the expectations.

Reviewers' comments:

Reviewer #1 (Remarks to the Author):

The authors have implemented substantial revisions addressing the original reviewers comments/suggestions, and improving on already compelling manuscript. The main change in the revised manuscript is the addition of (numerical) calculation of dynamic fault weakening by thermal pressurization (TP) of pore fluid and corresponding evolution of the bulk fault temperature under constant rate of slip using the full set of temperature-dependent thermodynamic properties of the pore fluid (water). The results of these calculations are used to address the buffer effect of super-critical water transition on the extent of the TP fault weakening, to confirm the lack of the flash heating weakening for moderate ambient pore pressure ($p_f \sim 25\text{ MPa}$, 45 MPa), in general accord with their experiments, and extrapolate the findings to the mid-crustal conditions ($p_f \sim 70\text{ MPa}$). This is a worthwhile effort that enriches the paper, and allows to extrapolate the experimental findings to wider range of conditions in seismogenic crust, however, the implementation and the result also raise some questions.

1. Regarding the actual fault model that is numerically solved for the imposed constant slip rate. In the "Problem formulation" of "Bulk fault temp. Model", lines 572-581 of the Methods, the authors state that they consider a finite (but thin) shear zone thickness "wsz", yet then on lines 591-593 they state that they consider "the slip-on-a-plane derivation by JR Rice". The latter implies that $wsz=0$. So which model of the sheared zone the authors use? The heat diffusion equation Eq. (7), lines 595-596, contains the shear heating term which in place of the slip rate " v " has to have the shear strain rate. If the "slip-on-a-plane" model is used then the heating term in Eq. (7) is missing the Dirac delta function in y , i.e. the shear strain rate distribution is v times the Dirac function. Please clarify.

2. lines 127-130: When discussing the result of the fault bulk temperature calculations the authors state that "reached temperatures are not high enough to reach the flash heating temperature". Given that the authors refer to Fig 5a here, which shows the fault bulk temperature, this does raise a question. The flash heating weakening is to be activated when the asperity flash temperature (I.e borne locally on a small asperity during its contact duration) reaches the critical value (1000 C). T_{asperity} should be given by T_{bulk} (Fig 5a) plus the flash increase dT_{flash} (like that in Fig 4a). The estimate of the latter would be similar to that in Fig 4, with the notable difference that the thermodynamic properties of water exchanging heat with asperity in Eq. (5) of the Methods need to be evaluated at T_{asperity} which reference level (at the start of the asperity contact slip) is no longer the room temperature but rather the elevated T_{bulk} of the fault.

3. When discussing the implications of their results for the crust at various depths (e.g. 2.5 km, 4.5 km, 7 km) in Figure 5 (should be labeled Figure 6), the authors show bulk temperature of shear heated fault originating from the same ambient level at zero slip (room temperature). Since the calculations are non-linear (due to the properties dependence on temperature), the results for the bulk temperature increase will depend on the ambient level, which is clearly should be changing with depth in the crust, no?

Some minor comments to help further improve the paper clarity.

1. The authors refer to the water latent heat L_w of the super-critical (and/or vapour) transition, line 79 of the main text and Eq. (5) of the Methods. Please provide (e.g. in the Supp. Mat. Table 3) the relevant value(s) used in the calculations/estimates.

2. As discussed by the authors, lines 89-90, and suggested by their Eq.(5), the flash increase of the asperity temperature follows solution of Rice [2006] for the dry case, dT_{flash} proportional to $\sqrt{t_{\text{contact}}} = \sqrt{\text{slip}/v}$. When examining Fig4a for the dry case, the dT_{flash} vs slip appears to have an inflection point near the terminal value of slip (which the $\sqrt{\text{slip}}$ dependence would not have)?

3. Further, if I use the quoted values for the rock properties in Table 2, the contact shear stress $\tau_c = 4.2$ Gpa (line 484), and $V = 1$ m/s, I am getting for the dry case of Eq. (5), $dT_{\text{flash}} = \tau_c V \sqrt{[\text{slip}/V] / (\rho c \sqrt{[\pi \kappa]})} = 3407$ C at slip =10 microns and 4818 C at slip =20 microns. Corresponding dT_{flash} values read from Fig 4a appear lower.
4. Line 448 of the Methods, expression for t_c has a typo, it should be $t_c = r/v$?
5. Line 473, what do the authors mean here by "Hertzian distribution of contact"
6. There are two Figures labeled "5" in the main text.
7. Table 3 of Supp. Mat. use "k" for both rock thermal diffusivity and permeability.

Reviewer #4 (Remarks to the Author):

The 2nd round review:

The authors have put substantial efforts into this revision and have provided detailed responses to all of my comments. I really appreciate that. In some sense, the paper was renovated, and now it reads much better. Again, the paper is of great interests to both experimental and geophysical communities. It would be suitable for publication in Nature Communications.

I still have a major concern which I think is quite crucial.

By the way, I think the highlight point of the paper (water phase transition controls earthquake rupture dynamics by T-buffering) is not straightforwardly supported by the main observations, which are as follows: 1) Large stress drops (weakening) were observed in dry and low P_f experiments but mostly attributed to flash heating, 2) High P_f experiments showed less stress drops and weakening where the phase transition plays an important role. Maybe the authors need to rephrase it.

Major comments (the three can be merged into one, all related to the latent heat):

- 1) My first major concern is about heat capacity which I also commented in the original version. The authors add a new figure in the supplement to show that heat capacity has potentially strong effect in a zone just beyond 22 MPa. Yes, heat capacities are extremely large in this zone (up to 1400 times the value at 1MPa). But I do not think this answers my original question. Here I suggest to look at the contour map of enthalpy ($h = c_p T + h_0$), which I think makes more sense when investigating the energy budget. There is also a jump in enthalpy beyond the liquid-to-gas phase transition, which is not reflected by c_p but by a h_0 jump (the latent heat). Therefore, it is not fair to judge the temperature buffering effect by just looking at c_p .
- 2) Line 96-100 and Figure 4: I agree that the asperity temperatures were buffered at short time (slip distance). But for longer time, the authors claimed that the flash temperature predicted is similar to the dry case (say > 500 C). How does your calculation make the jump from the liquid to the gas phase? I checked your equations, which indeed include the latent heat term. I trust your computation but in your supplementary table 2, you seem to forget the latent heat (L_w). From the latent heat, one would expect a heat barrier (or a gentler rise) in the temperature evolution curve. However, in your Figure 4 and extended data, the slip-temperature curves show very sharp rise upon the phase transition. By the way, some of your equations do not show subscripts. Fluid properties are P and T dependent, but you forget to mention the local pore pressure used (the same as P_f ??).
- 3) The same problem exists for the bulk temperature simulations (Figures 5 and 6). You did not include the latent heat in your energy balance equations.

Minor ones:

Line 22 abstract: incomplete sentence.

Line 81: "assumption" - assuming

Line 82: what is "FH velocity" (not introduced)? The sentence "the thermal diffusion length is close to the asperity size at FH velocity" is not clear to me. Is thermal diffusion length or asperity size dependent on FH velocity?

Line 113: vaporisation - vaporization.

Line 120: "TP" is not clear here. Does it mean "vaporization" or "fluid pressurization" in general?

Line 131-133: The tempratures predicted for high Pf are indeed lower than those for 1 MPa. But this does not suggest what you claimed here.

Line 292: This is Figure 6. Is not it?

All the revision texts need to checked: I am not a Native English speaker, but I can tell that "all the revision texts" are poorly written compared with the original text. Maybe the seniors can help with that.

Reviewer #1 (Remarks to the Author):

The authors have implemented substantial revisions addressing the original reviewers comments/suggestions, and improving on already compelling manuscript. The main change in the revised manuscript is the addition of (numerical) calculation of dynamic fault weakening by thermal pressurization (TP) of pore fluid and corresponding evolution of the bulk fault temperature under constant rate of slip using the full set of temperature-dependent thermodynamic properties of the pore fluid (water). The results of these calculations are used to address the buffer effect of super—critical water transition on the extent of the TP fault weakening, to confirm the lack of the flash heating weakening for moderate ambient pore pressure ($p_f \sim 25\text{MPa}$, 45MPa), in general accord with their experiments, and extrapolate the findings to the mid-crustal conditions ($p_f \sim 70\text{MPa}$). This is a worthwhile effort that enriches the paper, and allows to extrapolate the experimental findings to wider range of conditions in seismogenic crust, however, the implementation and the result also raise some questions.

#1- We thank the reviewer for the extremely pertinent reviews. We have tried to consider all the remarks that have been made. We hope the revised manuscript meets the expectations.

1. Regarding the actual fault model that is numerically solved for the imposed constant slip rate. In the “Problem formulation” of “Bulk fault temp. Model”, lines 572-581 of the Methods, the authors state that they consider a finite (but thin) shear zone thickness “ w_{sz} ”, yet then on lines 591—593 they state that they consider “the slip-on-a-plane derivation by JR Rice”. The latter implies that $w_{sz}=0$. So which model of the sheared zone the authors use? The heat diffusion equation Eq. (7), lines 595-596, contains the shear heating term which in place of the slip rate “ v ” has to have the shear strain rate. If the “slip-on-a-plane” model is used then the heating term in Eq. (7) is missing the Dirac delta function in y , i.e. the shear strain rate distribution is v times the Dirac function. Please clarify.

#2- Thank you for the remark. This was a mistake made when modifying the initial manuscript. We are not using anymore the slip-on-a-plane derivation for thermal pressurisation analysis. Here, we use a numerical model for a thin shear zone of finite thickness. Therefore, we have modified the text in lines 588-590. Equation (7) has also been corrected since in our model the heating term contains the shear strain rate: v/w_{sz} . The corrections are as follows:

“The fault is sheared at a constant shearing rate v over a thin slip zone of thickness w_{sz} ^{2, 15} where the temperature, and fluid pressure increase with shear loading. Note that, in the manuscript results presented are for a shear zone thickness of $5\ \mu\text{m}$.”

And

“

$$\frac{\partial T}{\partial t} = \frac{1}{(\rho_b \cdot c_b)} \cdot \mu_0 \cdot (\sigma_n - p_f) \cdot \left(\frac{v}{w_{sz}}\right) + \alpha_{th} \cdot \frac{\partial^2 T}{\partial y^2}$$

(7) “

2. lines 127-130: When discussing the result of the fault bulk temperature calculations the authors state that “reached temperatures are not high enough to reach the flash heating temperature”. Given that the authors refer to Fig 5a here, which shows the fault bulk temperature, this does raise a question. The flash heating weakening is to be activated when the asperity flash temperature (i.e borne locally on a small asperity during its contact duration) reaches the critical value (1000 C). T_{asperity} should be given by T_{bulk} (Fig 5a) plus the flash increase dT_{flash} (like that in Fig 4a). The estimate of the latter would be similar to that in Fig 4, with the notable difference that the thermodynamic properties of water exchanging heat with asperity in Eq. (5) of the Methods need to be evaluated at T_{asperity} which reference level (at the start of the asperity contact slip) is no longer the room temperature but rather the elevated T_{bulk} of the fault.

#3- We agree with the reviewer that the discussion of lines 127-130 was rather ambiguous. In fact, the heating mechanisms at the contacts and of the bulk fault should operate at very different spatial and temporal scales. Therefore, the interactions between the two mechanisms remain rather complex. We therefore prefer not to comment on this interaction directly. So we have modified the text in lines 133-140 as follows:

“Under drained conditions, we observed that the reached temperatures (which are a maximum estimation of the possible temperature in the bulk fault since the shear stress for heat generation is taken as the static fault’s shear strength of our experiments) remained below rock’s thermal degradation temperature (~ 1000 °C) even for slips larger than the maximum slip observed in the experiments (~ 250 μm) (Fig. 5a). This observation is in agreement with our microstructural observations, since melting was not pervasive over the sample surface but was localized at asperity scale (Fig. 3c and Supplementary Fig. 4c), as predicted by flash heating theory.”

3. When discussing the implications of their results for the crust at various depths (e.g. 2.5 km, 4.5 km, 7 km) in Figure 5 (should be labeled Figure 6), the authors show bulk temperature of shear heated fault originating from the same ambient level at zero slip (room temperature). Since the calculations are non—linear (due to the properties dependence on temperature), the results for the bulk temperature increase will depend on the ambient level, which is clearly should be changing with depth in the crust, no?

#4- We completely agree with the reviewer. We have therefore added a geothermal gradient of 30 °C/km in our calculations for the depth extrapolation. The new calculations are presented in the revised manuscript Figure. 6. Note that even when including a temperature evolution with depth, the reached temperatures are higher but the observed trends remain valid. At depth, thermal decomposition temperatures can be reached for the overall bulk fault for long slip distances. We have modified the methods to describe the addition of the geothermal gradient in line 158:

“The extrapolation was done for a mean stress equal to the lithostatic overburden gradient of 27 MPa.km⁻¹, a hydrostatic fluid pressure rise of 10 MPa.km⁻¹, a geothermal gradient of 30 °C.km⁻¹ (ref.²⁷), and an initial friction of 0.7 (e.g. Methods).”

And line 662:

“Extrapolation to upper crustal depths

In order to extrapolate the model to upper crustal depths, we substituted the experimental stress used in the heat source term of (eq. 7) for a mean stress taken as the lithostatic overburden gradient of 27 MPa.km⁻¹ with a fluid pressure gradient of 10 MPa.km⁻¹, an initial friction of 0.7, and a geothermal gradient of 30 °C.km⁻¹ (ref.²⁷). The results of this extrapolation are presented in Fig. 6 and commented in the main text.”

Some minor comments to help further improve the paper clarity.

1. The authors refer to the water latent heat L_w of the super-critical (and/or vapour) transition, line 79 of the main text and Eq. (5) of the Methods. Please provide (e.g. in the Supp. Mat. Table 3) the relevant value(s) used in the calculations/estimates.

#5- We agree with the reviewer. All our equations include the latent heat and so, the complete enthalpy evolution with temperature. Nevertheless, the table did not show the water's latent heat of vaporization because it is temperature and pressure dependent. We have therefore presented the evolution of water's latent heat (and its value) which acts as a heat barrier in new Figure. 4.e and Supplementary Figure 6.e, j.

2. As discussed by the authors, lines 89-90, and suggested by their Eq.(5), the flash increase of the asperity temperature follows solution of Rice [2006] for the dry case, dT_{flash} proportional to $\sqrt{t_{\text{contact}}} = \sqrt{\text{slip}/v}$. When examining Fig4a for the dry case, the dT_{flash} vs slip appears to have an inflection point near the terminal value of slip (which the $\sqrt{\text{slip}}$ dependence would not have)?

#6- Thank you for the remark, we are very grateful that you figured out this issue. In fact, we had a mistake when computing the flash temperatures and presenting the results of temperature Vs. slip. We have corrected our calculations and the corrected results are presented in Figure 4 of the revised manuscript.

3. Further, if I use the quoted values for the rock properties in Table 2, the contact shear stress $\tau_c = 4.2$ Gpa (line 484), and $V = 1$ m/s, I am getting for the dry case of Eq. (5), $dT_{\text{flash}} = \tau_c V \sqrt{\text{slip}/V} / (\rho c \sqrt{\pi \kappa}) = 3407$ C at slip =10 microns and 4818 C at slip =20 microns. Corresponding dT_{flash} values read from Fig 4a appear lower.

#7- Thanks again for the remark, this problem arose from the same mistake described in reply **#6**. We have revised all our calculations and corrected this problem.

4. Line 448 of the Methods, expression for t_c has a typo, it should be $t_c = r/v$?

#8- Yes thank you. This has been modified in the manuscript.

5. Line 473, what do the authors mean here by “Hertzian distribution of contact”

#9- Thank you for the remark. This has been modified in the manuscript. The idea is to say that the contact height is equal to the contact radius for a spherical contact.

6. There are two Figures labeled “5” in the main text.

#10- Yes thank you. This has been modified in the manuscript.

7. Table 3 of Supp. Mat. use “k” for both rock thermal diffusivity and permeability.

#11- Yes thank you. This has been modified in supplementary material to use κ (kappa) for thermal diffusivity.

Reviewer #4 (Remarks to the Author):

The 2nd round review:

The authors have put substantial efforts into this revision and have provided detailed responses to all of my comments. I really appreciate that. In some sense, the paper was renovated, and now it reads much better. Again, the paper is of great interests to both experimental and geophysical communities. It would be suitable for publication in Nature Communications.

#12- We thank the reviewer for the very constructive reviews. We hope the revised manuscript meets the expectations.

I still have a major concern which I think is quite crucial.

By the way, I think the highlight point of the paper (water phase transition controls earthquake rupture dynamics by T-buffering) is not straightforwardly supported by the main observations, which are as follows: 1) Large stress drops (weakening) were observed in dry and lowPf experiments but mostly attributed to flash heating, 2) HighPf experiments showed less stress drops and weakening where the phase transition plays an important role. Maybe the authors need to rephrase it.

#13- Thank you for the remark, nevertheless, the vaporization phase transition allows for the flash heating mechanism to operate (as demonstrated both by Experiments and the flash temperature model). Thus, vaporization allows dynamic flash weakening in low pore fluid pressure conditions. On the other hand, the supercritical phase transition inhibits such mechanism therefore controlling dynamic weakening under high pore fluid pressure. For these reasons we stand by the highlight point of our paper. In order to clarify, we have modified the manuscript in line 102-104 as follows:

“In this case, vaporization of water enhanced shear heating at contacts and allowed FH of asperities for slip velocities larger than $\sim 10 \text{ cm}\cdot\text{s}^{-1}$, as also observed in dry conditions.”

Major comments (the three can be merged into one, all related to the latent heat):

1) My first major concern is about heat capacity which I also commented in the original version. The authors add a new figure in the supplement to show that heat capacity has potentially strong effect in a zone just beyond 22 MPa. Yes, heat capacities are extremely large in this zone (up to 1400 times the value at 1MPa). But I do not think this answers my original question. Here I suggest to look at the contour map of enthalpy ($h = c_p \cdot T + h_0$), which I think makes more sense when investigating the energy budget. There is also a jump in enthalpy beyond the liquid-to-gas phase transition, which is not reflected by c_p but by a h_0 jump (the latent heat). Therefore, it is not fair to judge the temperature buffering effect by just looking at c_p .

#13- We partially agree with the review. Our equations include the latent heat and so, the complete enthalpy evolution with temperature. Nevertheless, we agree that table 2 did not

show the water's latent heat of vaporization because it is temperature and pressure dependent. We have therefore presented the evolution of water's latent heat in new Figure. 4.e. and Supplementary Figure 6.e, j. In addition, we have modified the text in lines 87 as follows:

“We include here the isobaric evolution of water's specific and latent heat (c_{pw} and L_w), as well as density (ρ_w) with temperature²⁴ (Fig. 4c, d; e, Methods, Extended Data Fig.7).”

And lines 106 – 108:

“This phase change requires a distributed amount of energy over a finite temperature range, opposed to the case of isothermal vaporization where L_w acts as a heat barrier. Therefore, the heat capacity of water increases by 1400% during the transition at $pf=25$ MPa (Fig. 4d) while the drop in density is smoother than in the case of vaporization²⁵ (Fig. 4c).”

2) Line 96-100 and Figure 4: I agree that the asperity temperatures were buffered at short time (slip distance). But for longer time, the authors claimed that the flash temperature predicted is similar to the dry case (say > 500C). How does your calculation make the jump from the liquid to the gas phase? I checked your equations, which indeed include the latent heat term. I trust your computation but in your supplementary table 2, you seem to forget the latent heat (L_w). From the latent heat, one would expect a heat barrier (or a gentler rise) in the temperature evolution curve. However, in your Figure 4 and extended data, the slip-temperature curves show very sharp rise upon the phase transition. By the way, some of your equations do not show subscripts. Fluid properties are P and T dependent, but your forget to mention the local pore pressure used (the same as P_f ??).

#14- Please refer to comment **#5** and **#13**. As presented in the revised Figure.4.e and Supplementary Figure 6.e, j. and as described in the modified text line 78-80, the latent heat of vaporization acts as a heat barrier. In order to account for a controlled phase transition, it would be possible to take into consideration the volumetric fraction of each phase. Nevertheless, taking the latent heat of vaporization as a heat barrier is a reasonable assumption since, under flash heating theory, the temperature rise is an extremely fast process. Modifications are as follows:

“In the presence of fluids, water cools asperities through heat capacity and latent heat (acting as a heat barrier) of a finite water volume surrounding the highly stressed asperity⁷ (Fig. 4b).”

Regarding the local pore pressure used, we did mention that we include the isobaric evolution of water's properties with temperature (line 86) and the pore pressures used are detailed in Figure 4. as labels.

3) The same problem exists for the bulk temperature simulations (Figures 5 and 6). You did not include the latent heat in your energy balance equations.

#15- Thank you for the remark. The latent heat does not appear in the heat equation because, as discussed by Chen et al. (2017) the latent heat (here L_w) being constant for pressures lower than that of the supercritical phase transition, it vanishes when deriving the energy conservation equation since $(\partial \rho_w \cdot h_w) / \partial T = (\partial \rho_w \cdot (c_w \cdot T + L_w)) / \partial T = \rho_w \cdot c_w$. where h_w is the enthalpy of water.

We have added a paragraph in the methods referring to this in order to clarify the manuscript in lines 625-627 as follows:

“Note that, as discussed by Chen et al.¹⁵, the latent heat (here L_w) being constant for pressures lower than that of the supercritical phase transition, it vanishes when deriving the energy conservation equation since $\frac{\partial \rho_w \cdot h_w}{\partial T} = \frac{\partial \rho_w \cdot (c_w \cdot T + L_w)}{\partial T} = \rho_w \cdot c_w$. where h_w is the enthalpy of water.”

Minor ones:

Line 22 abstract: incomplete sentence.

#16- The sentence in the abstract is now complete. It refers to the effect described in the previous sentence. We have modified the text to clarify.

Line 81: "assumption" - assuming

#17- Here the assumption refers to the main hypothesis we explain in the sentence just above; we have modified the text.

Line 82: what is "FH velocity" (not introduced)? The sentence "the thermal diffusion length is close to the asperity size at FH velocity" is not clear to me. Is thermal diffusion length or asperity size dependent on FH velocity?

#18- Thanks, for the comment. This has been corrected in lines 84 as follows:

“(1) the thermal diffusion length ($\sqrt{\pi \cdot \kappa \cdot t}$ with κ the thermal diffusivity and t the heating time) is close to the asperity size at FH velocities (commonly admitted as $\sim 10 \text{ cm} \cdot \text{s}^{-1}$ (refs.²⁻⁷)).”

Line 113: vaporisation - vaporization.

#19- Yes thank you. This has been corrected in the revised manuscript.

Line 120: "TP" is not clear here. Does it mean "vaporization" or "fluid pressurization" in general?

#20- We refer to fluid pressurization. To clarify the text, we added the following in the revised manuscript line 125:

“TP due to fluid pressurization could then be a candidate to explain the slightly lower dynamic friction values observed at Low_{pf} while FH remains the dominant weakening mechanism.”

Line 131-133: The temperatures predicted for high Pf are indeed lower than those for 1 MPa. But this does not suggest what you claimed here.

#21- Please refer to response **#3** for reviewer 1.

Line 292: This is Figure 6. Is not it?

#22- Yes thank you, this has been corrected in the revised manuscript.

All the revision texts need to be checked: I am not a Native English speaker, but I can tell that "all the revision texts" are poorly written compared with the original text. Maybe the seniors can help with that.

#23- Thank you for the pertinent comments, we have reviewed the language problems, improved the manuscript, the models, figures, methods and supplementary information. We hope it meets the expectations.

Reviewers' comments:

Reviewer #1 (Remarks to the Author):

The authors have addressed the queries raised in the previous rounds of reviews. I am happy to recommend the manuscript for publication in Nature Comm.

Reviewer #4 (Remarks to the Author):

The 3rd round review:

In my second round of review, I asked for details of the implementation of water phase transition into the computation of both flash temperature (Figure 4) and bulk thermal pressurization (Figures 5 and 6). This is my only major concern in my last round of review. In the revised manuscript, the authors have 1) added a new column in Figure 4 showing latent heat as a function of temperature (Figure 4e) and 2) explained in lines 625-627 how they include phase transition in modeling bulk thermal pressurization. I think the ways they treated phase transition in both cases are problematic (see my arguments below).

Flash temperature calculation at Pf of 1 MPa:

[0-9um slip distance]: The authors argued that the liquid water and heated asperities are in thermal equilibrium which can therefore inhibit flash heating. I agree with this point ("0-9um slip distance").

[10-20 um slip distance]: Pore water becomes vapor which has low density and low heating capacity, so the sample became nominally "dry" and flash heating dominated the weakening. I also agree.

[Around 9 um]: Phase transition absorbs heat so there should be a temperature buffering effect in the temperature evolution result. However, what is showed in figure 4 between 9-10um slip is an abrupt temperature rise upon the occurrence of vaporization (NO buffering effect). If I understand correctly (from the added Figure 4e), the author treated latent heat as a continuous heat sink. Actually, phase transition only occurs at the boiling temperature. Since the present model does not consider pore pressure change around the asperities, the flash temperature should maintain constant for some time (slip distance) due to the vaporization. In this period, the authors can use " $\tau_{slip} = \text{accumulated latent heat of the vaporized portion of fluid}$ ".

Bulk thermal pressurization with phase transition:

The equation given in lines 625-627 is wrong. I went through the Appendix A given by Chen et al., 2017 (in particular A1-A7). I agree that when phase transition occurs, an extra term related to vapor (or liquid) saturation (their Sv or Sl) should be included. This does not matter if latent heat is constant or not, but is decided by Gibbs' phase rule. Therefore, one term is missing in equation (7).

I am sorry for having not recognized this earlier. I hope I am just too skeptical and totally wrong. If I am right, I am not sure if this issue will change the final conclusion or not, but it will definitely change a few curves in the main figures of the paper.

Considering the precious dataset and new concept (theory) proposed in the paper, I still strongly think this work would be suitable for publication in Nature Communications after clarifying this point.

The 3rd round review:

In my second round of review, I asked for details of the implementation of water phase transition into the computation of both flash temperature (Figure 4) and bulk thermal pressurization (Figures 5 and 6). This is my only major concern in my last round of review. In the revised manuscript, the authors have 1) added a new column in Figure 4 showing latent heat as a function of temperature (Figure 4e) and 2) explained in lines 625-627 how they include phase transition in modeling bulk thermal pressurization. I think the ways they treated phase transition in both cases are problematic (see my arguments below).

Flash temperature calculation at Pf of 1 MPa:

[0-9um slip distance]: The authors argued that the liquid water and heated asperities are in thermal equilibrium which can therefore inhibit flash heating. I agree with this point ("0-9um slip distance").

[10-20 um slip distance]: Pore water becomes vapor which has low density and low heating capacity, so the sample became nominally "dry" and flash heating dominated the weakening. I also agree.

[Around 9 um]: Phase transition absorbs heat so there should be a temperature buffering effect in the temperature evolution result. However, what is showed in figure 4 between 9-10um slip is an abrupt temperature rise upon the occurrence of vaporization (NO buffering effect). If I understand correctly (from the added Figure 4e), the author treated latent heat as a continuous heat sink.

We understand the reviewer's remark and the origin of this confusion. By definition, the flash temperature does not consider the kinetics of the heating process but rather the maximum transient temperature that can be reached. In our model, temperature is affected by a maximum buffering effect due to the water's phase transition. Consequently, the notion of the reaction kinetics does not apply to our model. We are rather computing the maximum temperature at asperities affected by the maximum buffering effect due to water cooling. If the kinetics of the phase transition were considered, the temperature should maintain constant for some time as stated by the reviewer. In that sense, we did not treat the latent heat as a continuous heat sink but rather as a local heat barrier. The latent heat represents the change in enthalpy (or energy per unit mass) needed to overcome the phase transition. In that sense, the final enthalpy of the system (represented by $h=cp*T+Lw$) is larger for the vapor phase than for the liquid phase, reason why Lw stays at its vapor value when temperatures are high enough to overcome the transition.

We have modified the manuscript in lines 91-93 in order to clarify the flash temperature model:

"Note that in this model we consider the maximum temperature that can be reached at asperities affected by the cooling effect of water⁷. Such temperature differs from the temperature history at asperities during slip. In that sense, no reaction kinetics is taken into account in this model."

Actually, phase transition only occurs at the boiling temperature. Since the present model does not consider pore pressure change around the asperities, the flash temperature should maintain constant for some time (slip distance) due to the vaporization. In this period, the authors can use " $T_{ao} * \text{slip} = \text{accumulated latent heat of the vaporized portion of fluid}$ ".

With regards to the suggestion of the reviewer that we use “ τ_{slip}^* = accumulated latent heat of the vaporized portion of fluid” we do not understand. A quick dimensional analysis yields an inconsistency in this equation as τ_{slip}^* has a unit of [MPa.s] = [J.m⁻²] and the latent heat would be expressed in [J.kg⁻¹] or, at best, accumulated latent heat per unit displacement would be expressed in [J.kg⁻¹.m⁻¹]. If there are still some obscure concepts about these calculations, we would be very glad if the reviewer can detail this point.

Bulk thermal pressurization with phase transition:

The equation given in lines 625-627 is wrong. I went through the Appendix A given by Chen et al., 2017 (in particular A1-A7). I agree that when phase transition occurs, an extra term related to vapor (or liquid) saturation (their S_v or S_l) should be included. This does not matter if latent heat is constant or not, but is decided by Gibbs' phase rule. Therefore, one term is missing in equation (7).

Thank you for the remark, again, we understand the reviewer concern. Nevertheless, in our model, we explicitly chose not to include the third term of appendix A of Chen et al. 2017, because here, we do not consider the kinetics of the vaporization transition. In fact, we consider that the water vaporization is an instantaneous reaction due to fast shear heating (the whole heating and weakening process occurs in ~500 microseconds in our experiments and calculations).

The main reasons for us not considering the kinetics of the reaction are as follows: (1) adding kinetics of the vaporization reaction would result in an added complexity that we do not want to add to this article. In fact, as stated in our previous two response letters to reviewers, and as shown by the text, this article highlights our new and original experimental results and the modelling has been added to support such experimental results. (2) the kinetics of vaporization are extremely difficult to constrain and should be the focus of more detailed modelling work which is far from the scope of this paper. And (3) the kinetics of the phase transition are only important in the vaporization domain while the kinetic term disappears in the supercritical domain. In nature, even under upper crustal stress conditions, the vaporization of fault water is not often expected. In fact, even for anthropogenic earthquakes, the fluid pressures should be close to or larger than the critical pressure (~22 MPa).

This being said, we understand that confusions could arise. We have therefore added the following paragraph in the methods section stating the limitations of our model in lines 669-676 as follows:

“The following limitations are noticeable:

- The kinetics of the vaporization reaction are not considered in this model. For details on some attempts to constrain such kinetics refer to Ref¹⁵. Instead, we have considered that the vaporization reaction is instantaneous and that the latent heat acts as a heat barrier. In the case of the supercritical transition, the terms concerning the kinetics of the reaction vanish^{15,16}.

- All heterogeneities that exist in fault zones¹ (in terms of, thermal, mechanical, and hydraulic properties normal and parallel to the fault plane) are neglected in this model.”

And in lines 630-631: “In addition, this model does not consider the kinetics of the vaporization transition but rather an instantaneous phase change when temperature is high enough to overcome vaporization.”

I am sorry for having not recognized this earlier. I hope I am just too skeptical and totally wrong. If I am right, I am not sure if this issue will change the final conclusion or not, but it will definitely change a few curves in the main figures of the paper.

Considering the precious dataset and new concept (theory) proposed in the paper, I still strongly think this work would be suitable for publication in Nature Communications after clarifying this point.

We would like to thank the reviewer for all his comments that have improved the manuscript. In addition, we would like to highlight the fact that adding further complexity to the models is far from the scope of this article. We strongly believe that adding further complexity to the models is of second order importance with respect to the experimental results presented.

REVIEWERS' COMMENTS:

Reviewer #4 (Remarks to the Author):

The revised manuscript has been improved and the authors address all of my concerns. I recommend this manuscript should be published in Nature Comm.

Just one suggestion: for the method of flash temperature calculation, clarification for the role of vaporization is necessary.